# The Early Paleozoic Tectonic Framework and Evolution of Northern West Qinling Orogen: By Zircon U-Pb Dating and Geochemistry of Rocks from Tianshui and Sunjiaxia

Zhen Liu [1,2] , Wei Xu [1,]*, Chunming Liu [2,]*, Yujia Xin [3] and Dezhi Huang [2]

1   Department of Geology and Architectural Engineering, Anhui Technical College of Industry and Economy, Hefei 230001, China; liu_zhen071@126.com
2   School of Geosciences and Info-Physics, Central South University, Changsha 410083, China; dzhuang_01@163.com
3   Institute of Geomechanics, Chinese Academy of Geological Sciences, Beijing 100081, China; xyjasd123@163.com
*   Correspondence: wei-xu@126.com (W.X.); liuchunming@csu.edu.cn (C.L.)

**Abstract:** The Tianshui-Sunjiaxia area is located in the connection zone of West Qinling Orogen and North Qilian Orogen, which could provide great insights into the amalgamation processes between the northern and southern blocks of China. Three subduction- and rift-related rocks gneissic granite from North Qilian arc-interarc belt (NQAI) granite and metabasalt from North Qinling back-arc basin (NQBA) are distinguished across the connection zone. The gneissic granite was generated by melts from older crustal materials of Longshan Group with the addition of a relatively juvenile basaltic source from the lower crust during the collision process. The Liwanxincun metabasalt reflects the mixing of the partial melting of the shallow asthenospheric mantle and the metasomatized mantle in a back-arc extension setting. The LA-ICP-MS zircon U-Pb dating of gneissic granite (068, 069) yields crystallization ages of $457.0 \pm 1.6$ Ma and $445.9 \pm 2.1$ Ma. The study area is divided into six tectonic units in Early Paleozoic time involving NQAI (Yanjiadian-Xinjie) continental arc, interarc rift basin (Maojiamo-Xiwali), continental arc (Chenjiahe-Wangjiacha); NQBA back-arc rift basin (Huluhe-Hongtubao), island arc and ophiolitic melange belt (North Qinling-Shangdan). A tectonic model is proposed in which the NQAI continental arc (Yanjiadian-Xinjie) might represent the early period of subduction of North Qilian Ocean (NQO) and the interarc rift is the product of the extension triggered by southward subduction of NQO. The ongoing subduction of NQO then leads to the formation of Chenjiahe-Wangjiacha continental arc, as well as the Hongtubao back-arc spreading ridge in NQBA back-arc basin (Huluhe). The tectonic evolution of the connection zone is closely associated with the closure of the North Qilian Ocean and North Qinling-Shangdan Ocean in the context of the convergence process of micro-continental blocks, including North China block, Longshan group and North Qinling Terrane.

**Keywords:** early Paleozoic; U-Pb zircon age; North Qilian Orogen; West Qinling Orogen; Central China; petrogenesis; tectonic setting; tectonic implication





## 1. Introduction

Qinling Orogenic Belt (QOB), which is a part of the Kunlun-Qilian-Qinling-Dabie orogenic system (Chinese Central Orogenic Belt), is a famous composite continental orogenic belt separating the northern and southern blocks of China. It has undergone complicated tectonic evolutionary processes and experienced multiple instances of separation, aggregation and collision of continental blocks since the late Proterozoic [1,2]. Western Qinling Orogenic Belt (WQOB) is the western extension of the North Qinling Orogenic Belt (NQOB), which is adjacent to the North Qilian Orogen (NQO) in the north and is connected to the East Kunlun Orogen (EKO) in the west (Figure 1). Of the whole tectonic evolution process,

the early Paleozoic is the most important episode for the WQOB and the NQO. During the early Paleozoic, in the region of WQOB and the NQO, multiple island arc, back-arc basins and oceanic (back-arc) basin subduction zones developed [3–5], which resulted in the present complex tectonic pattern of WQOB and the NQO. Thus, fully understanding the tectonic framework and evolution of northern West Qinling-South Qilian Orogen during the early Paleozoic is critical to recognize the amalgamation processes of the northern and southern blocks of China, and accordingly, the tectonic evolution of the Chinese mainland.

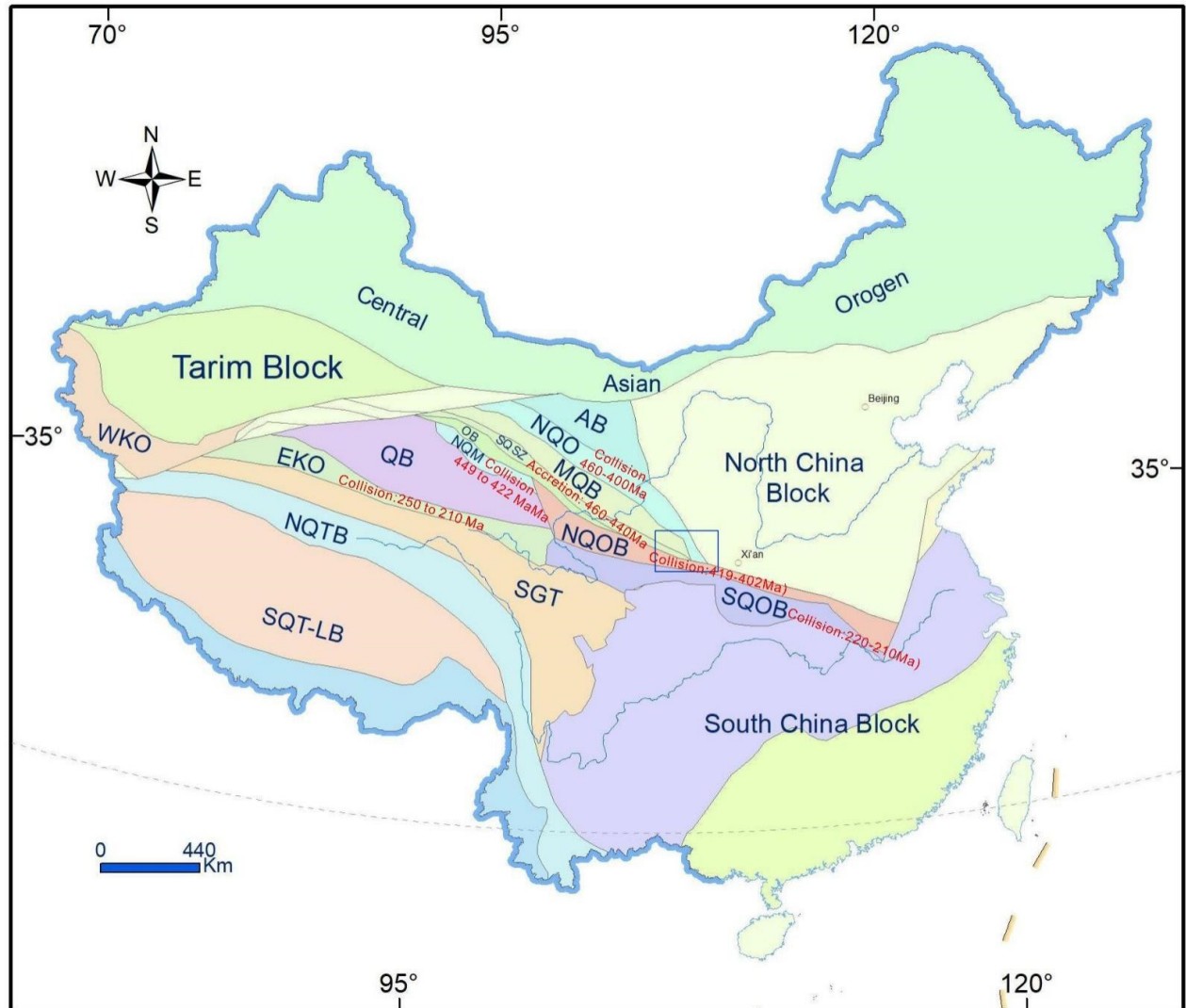

**Figure 1.** A simplified tectonic division map of China modified after Wang et al. (2013) [6] and components of the Central China Orogenic System and its adjacent blocks, marking the location of the study area. Abbreviations: NQO: North Qilian Orogen (collision: 460–400 Ma [3]); SQSZ: South Qilian Suture Zone (accretion: 460–440 Ma [7]); NQM: North Qaidam UHPM (collision: 449–422 Ma [8]); NQOB: North Qinling Orogenic Belt (collision: 419–402 Ma [9]); SQOB: South Qinling Orogenic Belt (collision: 220–210 Ma [4]); EKO: East Kunlun Orogen (collision: 250–210 Ma [10]); WKO: West Kunlun Orogen. AB: Alxa Block; MQB: Middle Qilian Block; QB: Qaidam Block; TB: Tarim Block; OB: Oulongbuluke Block; SGT: Songpan-Ganzi Terrane; SQT-LB: South Qiangtang and Lashan Blocks; NQTB: North Qiangtang Block.

The widespread outcrops of the early Paleozoic intrusions and Precambrian basement rocks in the region from the northern WQOB to the South Qilian Suture Zone (SQSZ) (Figure 1) provide us with the objects of study to probe into the tectonic framework of the region during the early Paleozoic. Many researchers have tried to reconstruct the

paleo-tectonic framework of this region in the post-Proterozoic by studying these rocks in different ways. By comprehensive analysis of the Middle Proterozoic metamorphic volcanic rocks in the WQOB and the adjacent areas, Zhang (1995) [11] considered that these rocks are rift-type volcanic rocks and suggested that several rift zones developed during the middle Proterozoic. By analyzing the geochemistry of the post-Sinian granites, Li (2008) [12] concluded that the granites in Tianshui area were formed under different tectonic settings and this district has different tectonic units. Pei et al. (2007) [13] considered that the Guanzizhen intermediate-basic igneous complex was formed in the island arc tectonic environment. Dong et al. (2011) [14] considered that Yanwan-Yinggezui ophiolites in Taibai area are E-MORB-type volcanic rocks formed in a back-arc basin setting, which reveals an early Paleozoic subduction belt along Weihe Fault.

However, the granites in this region correspond to different tectonic environments and are mixed in distribution and the study is still relatively weak. In the area from Sunjiaxia to Tianshui of the connection zone of the WQOB and eastern edge of NQO, multi-stage granite and granite gneiss from Sinian are presented in an exposed manner. Wangdian tonalites show characteristics of island arc, which reveal that an ocean basin was developed at the eastern end of the Early Paleozoic Qilian orogenic belt [15]. The distribution of litho-tectonic units in this area remains uncertain.

In this study, we provide integrated results of whole rock elements and Sr-Nd isotopes for the Kangjiali (047) gneissic granite, the Liwanxincun (045) metabasalt and new zircon U-Pb ages of Sunjiaxia (068, 069) gneissic granite across the connection zone (Figure 2) to constrain the emplacement ages and petrogenesis of these plutons.

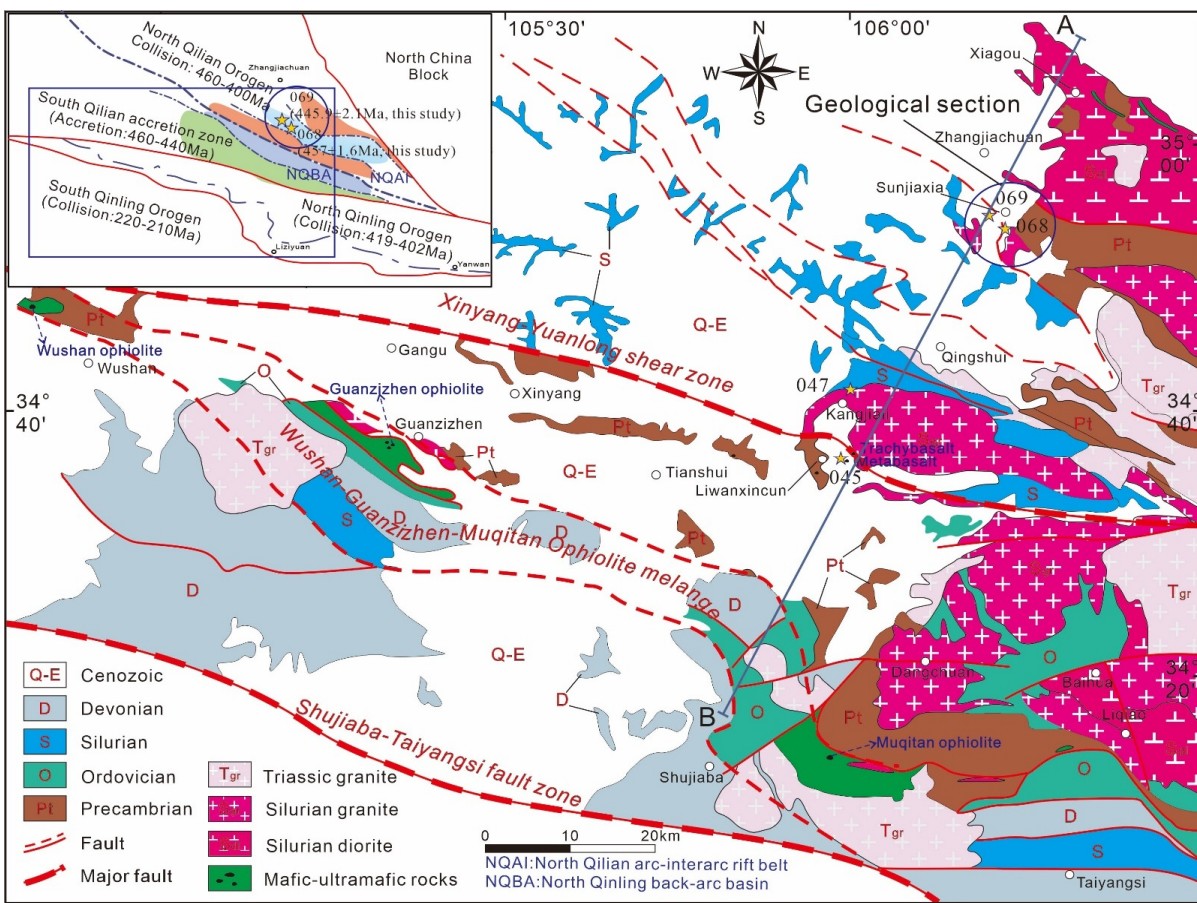

**Figure 2.** Simplified geological map of the Tianshui-Sunjiaxia area; modified after Pei et al. (2009) [16]. The blue line shows the geological section across the connection zone between the North Qilian Orogen and the West Qinling Orogen.

Combined with the previous studies on ophiolites, island arc- and back-arc-related magmatic rocks, we attempt to divide the rock tectonic units of the early Paleozoic orogenic belt in the connection zone and determine the evolution history of each tectonic unit. Finally, we discuss the tectonic implications in the context of the convergence process of multiple micro continental blocks and oceanic basins along the northern margin of the eastern Gondwana in Early Paleozoic time.

## 2. Geological Setting

The study area straddles the southern margin of North China Block and northern West Qinling Orogen. The strata outcropped in the region mainly include Proterozoic Qinling Complex, Cambrian to Ordovician volcano-sedimentary sequences, Silurian Taiyangsi Formation and Devonian Dacaotan Formation (Figure 2). Qinling Complex is mainly composed of garnet-sillimanite gneisses, amphibole-bearing two-pyroxene granulites, marbles and amphibolites [14,17]. The Cambrian to Ordovician strata include the Liziyuan and Caotangou Formation. Liziyuan Formation is mainly composed of green schist, siltaceous slate, sericite-quartz schist, plagioclase amphibole schist, feldspar sandstone and andesite. Caotangou Formation is a set of metamorphic volcanic-sedimentary rocks, mainly composed of biotite quartz schist, metamorphic sandstone, sandstone slate and phyllite, with a small amount of tuffaceous sandstone and andesite. Silurian Taiyangsi Formation is mainly composed of sericite-chlorite quartz schists and metaquartz sandstone. Devonian Dacaotan Formation is a set of metamorphic terrestrial clastic rocks, mainly including conglomerate, quartz sandstone, feldspathic quartz sandstone, siltstone and siltaceous mudstone.

Silurian granites are developed in this area. Xinyang-Yuanlong shear zone passes through the study area (Figure 3). To the south of the study area, there are several ophiolite sites suites distributed along Wushan-Guanzizhen-Muqitan in a nearly west-east direction. The Guanzizhen ophiolite consists mainly of greenschist facies, gabbros, pyroxenites, diorites and plagiogranites, which is considered as a typical N-MORB-type ophiolite [14,18] and was formed before 471 Ma for this suite [13]. The Wushan and Yanwan ophiolites show E-MORB geochemical features and were formed during 518–440 Ma [14,19].

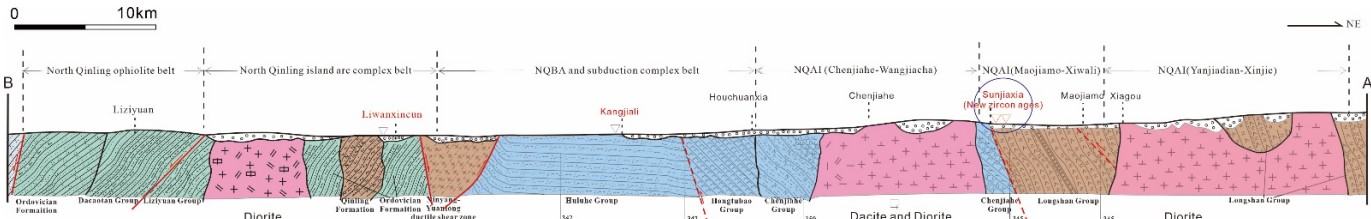

**Figure 3.** The geological section crossing the study area, with occurrence of representative structures and strata.

## 3. Sample Description and Analytical Methods

### 3.1. Whole-Rock Major and Trace Element Analyses

Whole-rock major and trace element compositions of 3 sets of samples were analyzed at the Central South China Supervision and Inspection Center of Mineral Resources, Ministry of Land and Resources, China. The fresh rocks were crushed to produce the powder of grain size less than 200 mesh. For the whole-rock major element analysis, the sample powders were then converted into glass fusion discs at 1000 °C by high-frequency melting furnace. Major elements were measured by XRF at Wuhan Institute of Geology and Resources, Wuhan, China. The analytical uncertainty is commonly less than 5%. For whole-rock trace element, the sample solutions which were produced by acid digestion were measured using ICP-MS at Wuhan Institute of Geology and Resources, Wuhan, China. The detailed digestion of samples followed the description by Liu et al. (2008) [20]. The analytical uncertainties for trace elements with concentrations more than 10 ppm and less than 10 ppm were within 5% and 10%, respectively.

### 3.2. Whole-Rock Sr and Nd Isotope Analyses

Representative samples from the Liwanxincun metabasalt and Kangjiali gneissic granite were selected and measured by MAT-261 mass spectrometer at Wuhan Institute of Geology and Resources, Wuhan, China. For Sr isotopic composition, standard NBS987 and NBS607 were used to monitor the technological process and the instruments, and the deviations were accordant with the recommended values by the certificate, with the errors within 0.015%. The measured average values of NBS feldspar standard were accordant with the recommended values, which is $65.46 \times 10^{-6}$ for Sr and $1.20048 \pm 52$ (2σ) for $^{87}Sr/^{86}Sr$. For Nd isotopic composition, the samples were dissolved by HF and $HClO_4$ to separate Sm and Nd by HDEHP coated on Teflon powder. Through isotope dilution method, abundances of Sm and Nd were determined by $^{149}Sm$- and $^{145}Nd$-enriched spikes. Nd ratios were normalized following a power law fractionation correction with $^{146}Nd/^{144}Nd = 0.7219$. The reproducibility of isotopic ratios was better than 0.005% and the precision for Sm and Nd concentrations was less than 0.5% at the two-sigma level.

### 3.3. Zircon U-Pb Dating

Sunjiaxia (068, 069) gneissic granite from the Longshan Formation was selected for zircon U-Pb dating. Zircons for U-Pb dating were separated using density and magnetic separation and then this was followed by hand-picking under a binocular microscope. Zircons were fixed in epoxy resin to polish to half thickness to expose their interior. The cathode luminescence (CL) images were taken using a JEOL scanning electron microscope to observe their internal textures and guide U-Pb dating. The zircon grains were analyzed using a laser ablation (LA)-MC-ICP-MS at Wuhan Institute of Geology and Resources, Wuhan, China. The external standards for Pb/U ratios and trace element concentrations are reference zircon GJ-1 and 91500, respectively. The common Pb correction and detailed analytical techniques followed procedures described by Liu et al. (2011) [21]. All age calculations and Concordia-diagram plots were conducted by ISOPLOT (ver. 3.0) [22]. The uncertainties for individual analyses were quoted at 1σ, whereas the errors for weighted mean ages were quoted at 2σ.

## 4. Results

### 4.1. Bulk-Rock Major Elements, Trace Elements and Sr-Nd-Hf Isotopes

Whole-rock major and trace element and Sr-Nd isotopic compositions of the studied samples are presented in Supplementary Tables S1 and S2.

#### 4.1.1. The Kangjiali (047) Gneissic Granites

The Kangjiali gneissic granite (047) has $SiO_2$ contents of 71.77–76.73 wt.%. The contents of $Al_2O_3$, CaO, $K_2O$, $Na_2O$, $Fe_2O_3$, MgO and $TiO_2$ are 9.79–12.60 wt.%, 0.55–1.86 wt.%, 1.88–3.74 wt.%, 0.64–2.16 wt.%, 0.53–1.69 wt.%, 1.74–2.46 wt.% and 0.53–0.63 wt.%, respectively. The rocks are rich in MgO ($Mg^{\#} = 39.93-46.89$) and $K_2O$ ($K_2O/Na_2O > 1$) and have moderate $K_2O + Na_2O$ contents of $3.32-4.66$ wt.%. On the classification diagram of $SiO_2$ versus $Na_2O + K_2O$ (TAS) (Figure 4), they fall in the fields from granite to granodiorite. Because of their chemical diversity, they fall into the peraluminous fields (Figure 5) and spread across the boundaries between medium calc-alkaline and high-K calc-alkaline fields (Figure 6).

These samples have REE contents of 148.13–215.40 ppm and show fractionated REE patterns characterized by LREE enrichment and HREE depletion with $La_N/Yb_N = 8.41–11.74$ (Figure 7, Table S1). They show evident negative Eu anomalies with $Eu/Eu^* = 0.61–0.66$ (Figure 7, Table S1). Kangjiali samples show obvious LREE/HREE fractionation with $(La/Sm)_N$ ratios of 3.79–4.71, $(La/Yb)_N$ ratios of 8.41–11.74 and $(Gd/Yb)_N$ ratios of 1.27–1.87 and weak Eu anomalies ($Eu/Eu^* = 0.91–0.93$). On the spider diagram (Figure 8), they are enriched in Rb, U and Th, and depleted in Nb, Ta, P and Ti. The Sr-Nd isotopic compositions of the samples are homogeneous. The whole-rock $^{87}Sr/^{86}Sr$ ratios of Kangjiali gneissic

granites range from 0.72258 to 0.75448; the $^{143}Nd/^{144}Nd$ ratios range from 0.511775 to 0.51188 (Table S2).

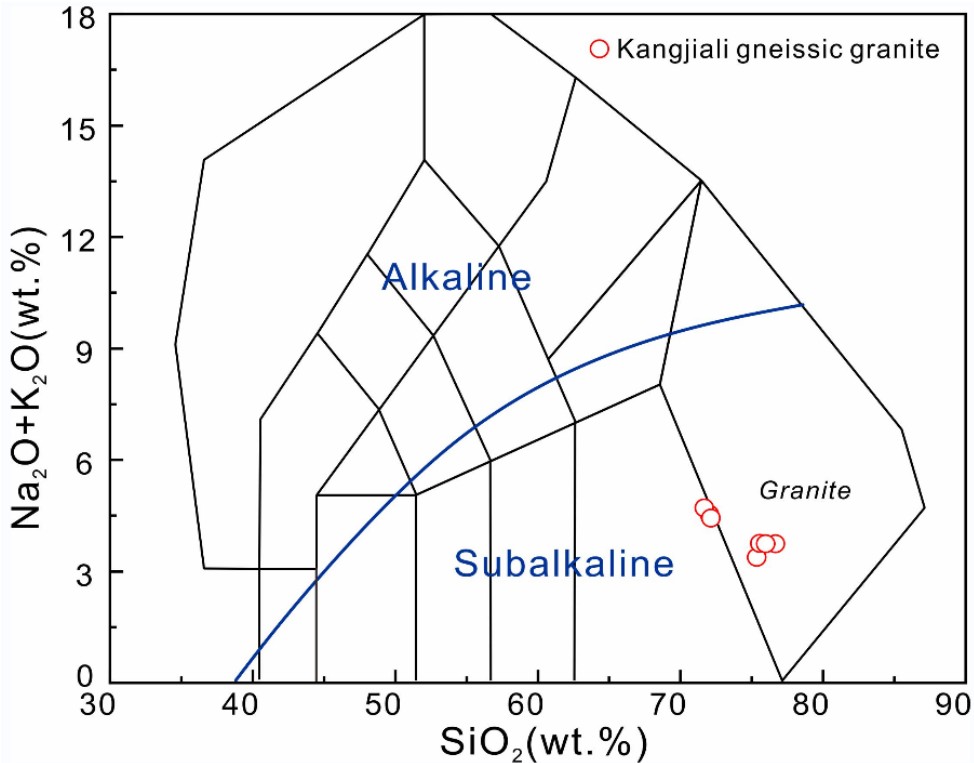

**Figure 4.** TAS diagram [23] of the Kangjiali gneissic granite.

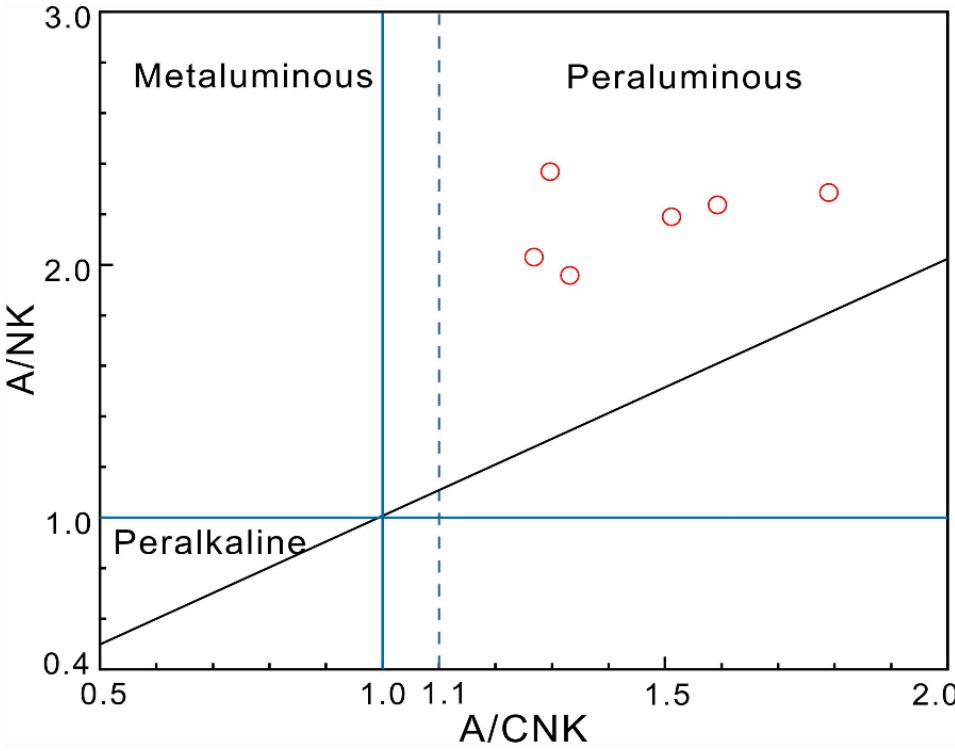

**Figure 5.** A/CNK vs. A/NK diagram [24] of the Kangjiali gneissic granite.

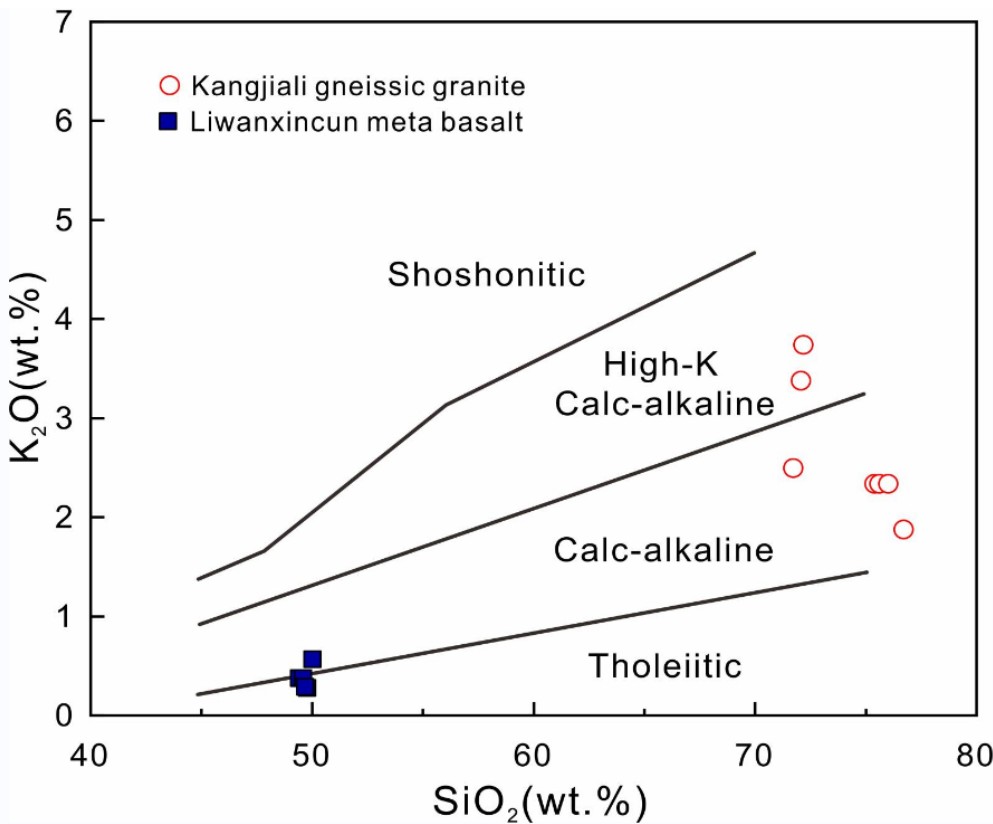

**Figure 6.** SiO$_2$ versus K$_2$O diagram [25] of the Kangjiali gneissic granite and Liwanxincun metabasalt.

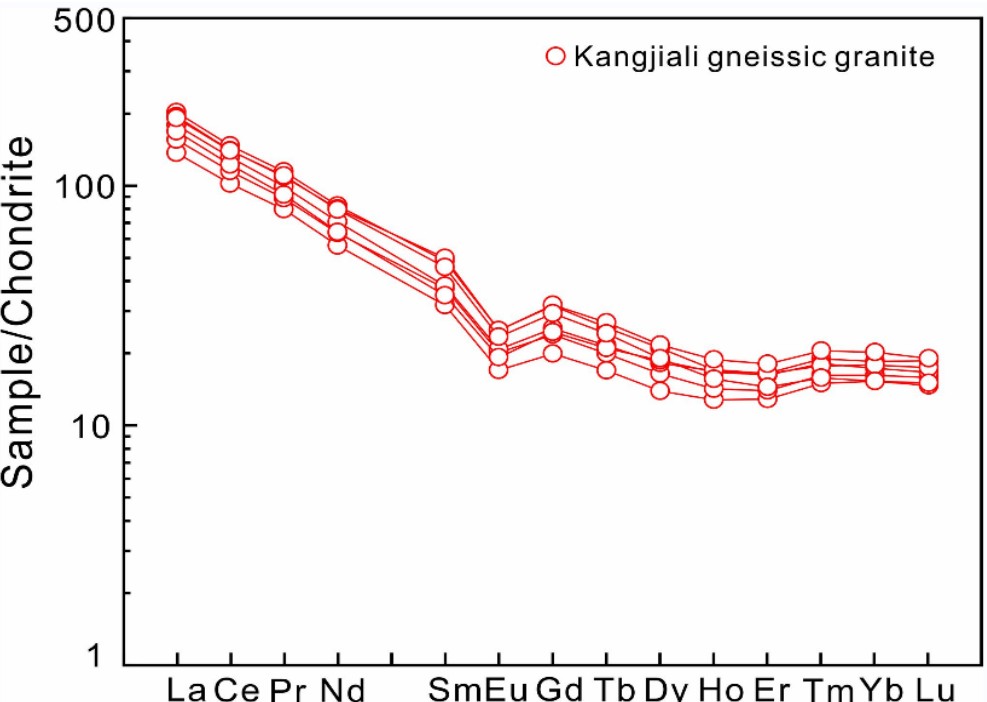

**Figure 7.** Chondrite-normalized rare element (REE) diagram of the Kangjiali gneissic granite. Chondrite data are from Sun and McDonough (1989) [26].

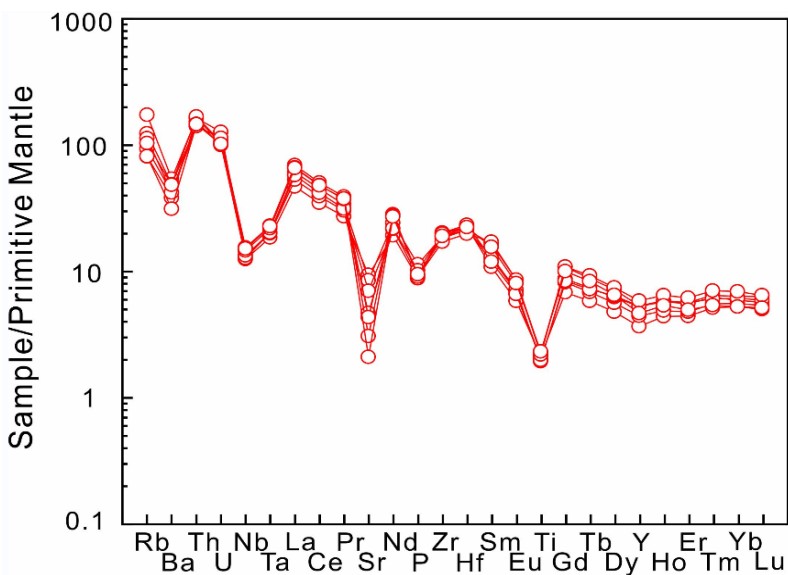

**Figure 8.** Primitive-mantle-normalized trace element diagram of the Kangjiali gneissic granite. Primitive mantle data are from Sun and McDonough (1989) [26].

### 4.1.2. The Liwanxincun (045) Metabasalt

The Liwanxincun metabasalt from the Hongtubao Formation has $SiO_2$ contents of 49.47–50.08 wt.%. The contents of $Al_2O_3$, CaO, $K_2O$, $Na_2O$, $FeO_t$, MgO and $TiO_2$ are 13.94–14.76 wt.%, 8.44–10.46 wt.%, 0.29–0.58 wt.%, 1.26–2.55 wt.%, 10.82–11.41 wt.%, 7.72–8.50 wt.% and 1.32–1.61 wt.%, respectively. The rocks are characterized by high Na ($Na_2O/K_2O$ ratios range from 0.12 to 0.46) and are rich in Al ($Al_2O_3$ from 13.94 wt.% to 14.76 wt.%). On the plot of $K_2O$ versus $SiO_2$, the rocks straddle the calc-alkaline and tholeiitic series (Figure 6).

The contents of REE are from $53.16 \times 10^{-6}$ to $62.51 \times 10^{-6}$. All samples display similar REE distribution patterns characterized by slight enrichment in LREEs with weak Eu anomalies (Eu/Eu* = 1.07–1.09) (Figure 9). The $(La/Sm)_N$ ratios range from 1.06 to 1.16, $(La/Yb)_N$ range from 1.62 to 1.94. On the spider diagram (Figure 10), they show slight enrichment in LILEs, (e.g., Rb and Ba) and depletion in high field strength element (HFSEs, e.g., Nb, Ta, Zr and Hf).

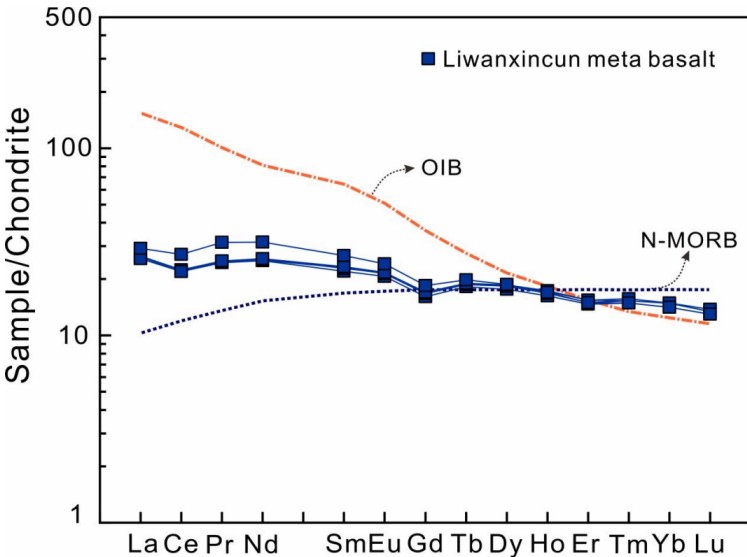

**Figure 9.** Chondrite-normalized rare element (REE) diagram of the Liwanxincun metabasalt. Chondrite data are from Sun and McDonough (1989) [26].

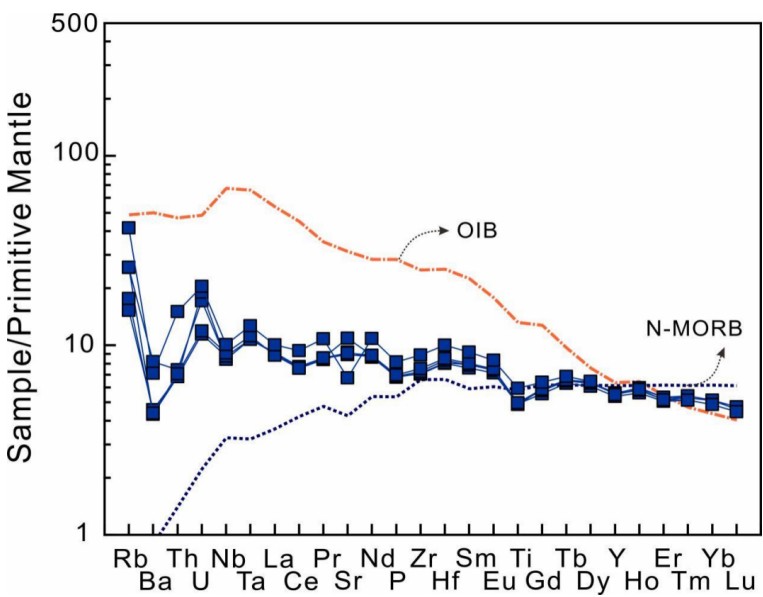

**Figure 10.** Primitive-mantle-normalized trace element diagram of the Liwanxincun metabasalt. PM data are from Sun and McDonough (1989) [26].

The whole-rock $^{87}Sr/^{86}Sr$ ratios range from 0.705318 to 0.706131, the $^{143}Nd/^{144}Nd$ ratios from 0.512094 to 0.512322.

### 4.2. Zircon U-Pb Ages

The zircon U-Pb analytical data are listed in Supplementary Table S3.

Zircon grains from Sunjiaxia (068) gneissic granite are euhedral and long prismatic. The grain sizes range from 100 μm to 160 μm in length with aspect ratios of about 1.5~3.0. Under cathodoluminescence, most of the grains exhibit oscillatory magmatic zoning as represented by zircon with no. 6 analyze point (Figure 11). Thirty-two analyses were conducted on 32 zircon grains. These zircons have variable Th/U ratios of 0.06–0.99 (Table S3). These grains yielded a weighted mean $^{206}Pb/^{238}U$ age of 457.0 ± 1.6 Ma (MSWD = 0.49; Figure 11), which is interpreted as the crystallization age of the diorite.

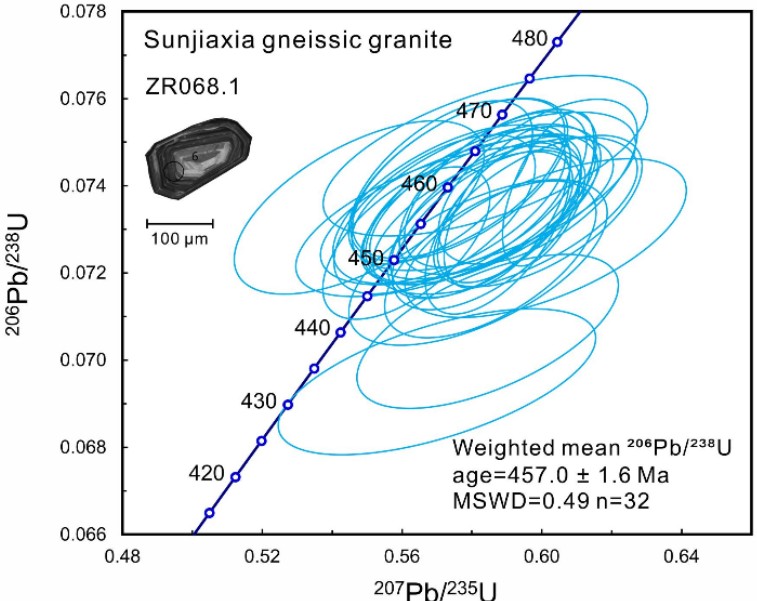

**Figure 11.** LA-ICP-MS U-Pb Concordia diagrams of zircons from the Sunjiaxia gneissic granite (ZR068.1).

Zircon grains from Sunjiaxia (069) gneissic granite are euhedral and exhibit oscillatory zoning (Figure 12). They show variable Th/U ratios ranging from 0.57 to 0.98. The five grains yield a weighted mean $^{206}Pb/^{238}U$ age of 445.9 ± 2.1 Ma (MSWD = 1.9; Figure 12), which is interpreted as the crystallization age of the gneissic granite.

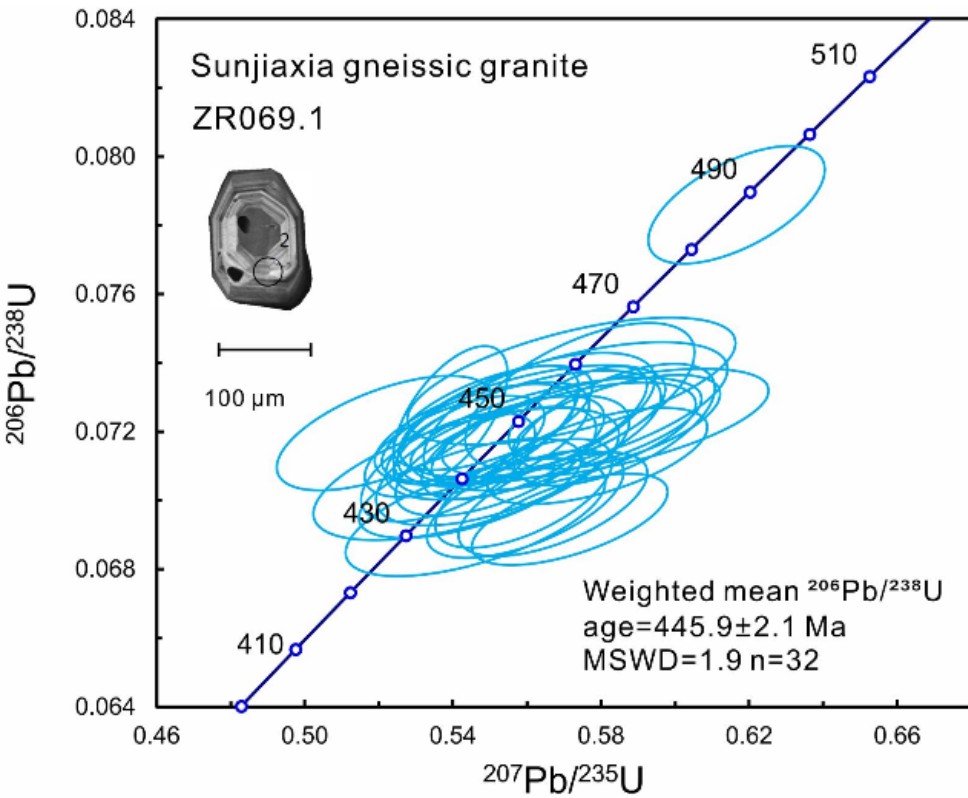

**Figure 12.** LA-ICP-MS U-Pb Concordia diagram of zircons from Sunjiaxia gneissic granite (ZR069.1).

## 5. Discussion

### 5.1. Petrogenesis and Magma Sources

#### 5.1.1. The Kangjiali Gneissic Granite

The Kangjiali gneissic granites are characterized by lack of Cr and Ni contents, LILEs enrichment and obvious negative Ba, Nb, Sr, P and Ti anomalies (Figure 8). These features are consistent with those of the typical arc-related volcanic rocks and granites in subduction zones in North Qilian Orogen [27,28].

The Kangjiali gneissic granites also show similar $Na_2O$ and $K_2O$ contents associated with high Sr and Al and high A/CNK (>1.1) signature of peraluminous S-type granites, which were generated by partial melting of metasedimentary rocks [29]. However, peraluminous granitoids can also form in subduction-related environments [30,31], generally exhibiting higher FeO, MgO, Sr and Al. Kangjiali gneissic granite falls into the arc-type rocks zone in the Ta vs. Yb diagram (Figure 13) and shows high Sr contents, which supports the subduction-related genesis. This is supported by volcanic arc feature in Rb/30-Hf-Ta*3 diagram (Figure 14).

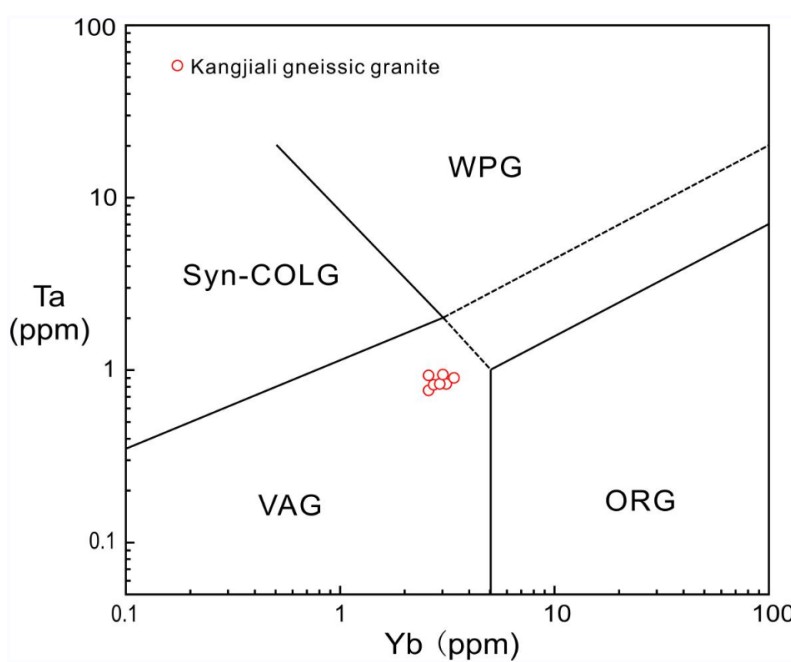

**Figure 13.** (Yb + Ta) versus Rb diagram and Yb versus Ta diagram [32] of the Kangjiali gneissic granite.

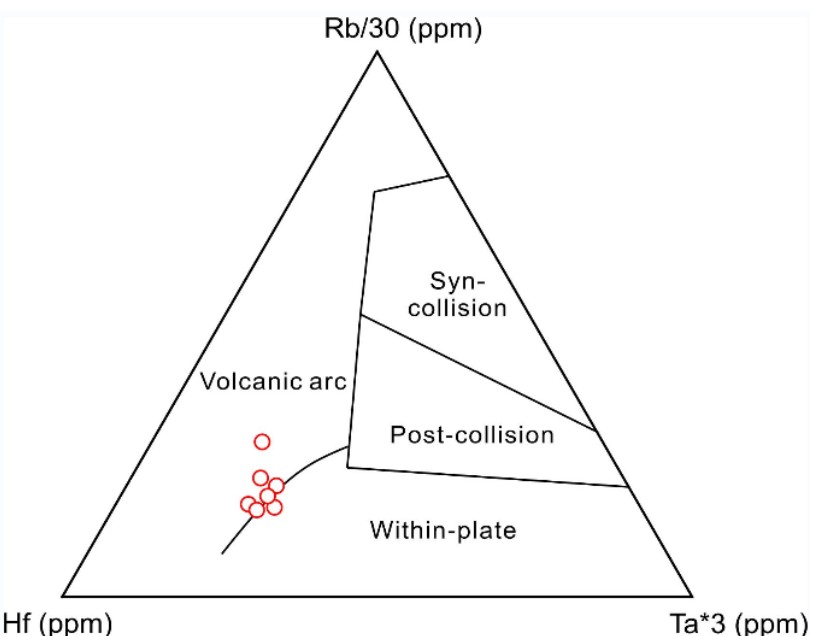

**Figure 14.** Hf–Rb/30-Ta*3 diagram [33] of the Kangjiali gneissic granite.

The source regions of the peraluminous volcanic rocks occurring in convergent margins have been variably attributed, including restite unmixing or peritectic assemblage entrainment [34], mafic lower crust [35,36] and subducting oceanic slab [37]. The absence of restitic/peritectic phases such as garnets and pyroxenes or restitic xenoliths precludes the restite unmixing and peritectic assemblage entrainment genesis. The evolved Sr–Nd isotopic compositions of these peraluminous granites is like the involvement of subducted oceanic slab melting in their genesis rather than mafic lower crust derivation.

The Kangjiali gneissic granite shows high $SiO_2$ (76.73–71.77 wt.%), moderate $Mg^{\#}$ values (39.93–46.89) and peraluminous characteristics (Figure 5), which is consistent with granites from partial melting of lower crustal materials. In addition, the Kangjiali gneissic

granite shows K-rich characteristics and falls in the area of continental crust in the $I_{Sr}$-$\varepsilon Nd(t)$ diagram (Figure 15), indicating that the source region is continental crust.

However, the Rb/Sr ratio of the Kangjiali gneissic granite is less than 0.9 (with an average of 0.73), which is inconsistent with the characteristic of S-type granite in typical continental crust source [38]. Ancient crustal materials are characterized by lower $\varepsilon Nd(t)$ and higher $I_{sr}(t)$, which may provide the source of peraluminous magmas. The Sr-Nd isotopic compositions of the granitic rocks show enriched features between the basaltic crust and ancient meta-basement materials in the upper crust (Figure 15). These features suggest that they were probably mainly derived from older crustal materials with the addition of a relatively juvenile basaltic source from the lower crust.

In addition, the isotopic features of Kangjiali gneissic granite are in good agreement with the trend of the isotopic mixing line between basaltic crust and Proterozoic basement of Longshan Group, demonstrating the source variation in their petrogenesis. The hybrid sources of the Kangjiali gneissic granite may possibly include ancient crustal materials from Longshan Group and newly underplated basaltic juvenile crust. Furthermore, these granitic rocks show exceedingly various $I_{Sr}(t)$ (Figure 15), indicating the newly underplated basaltic magma may incorporate subduction-related fluids or derive from source region metasomatized mantle by slab-derived fluids. Thus, we suggest that the Kangjiali peraluminous granite was derived from ancient crustal source materials and was significantly influenced by mantle-derived subduction-related magma.

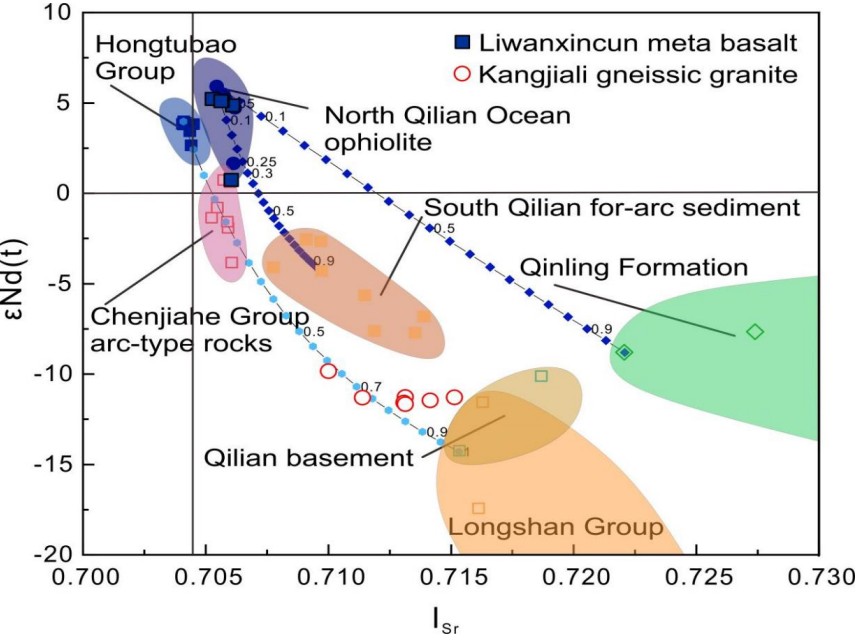

**Figure 15.** $\varepsilon Nd(t)$ versus $I_{Sr}$ diagram of the Kangjiali gneissic granite and the Liwanxincun metabasalt. The initial isotopic composition data are calculated at t = 443.8 Ma according to the U-Pb age of Changgouhe dioritic gneiss [39]. Data sources: North Qilian Ocean ophiolites from Hou et al. (2006) [40]; Hongtubao MORB-type basalts from Hu. (2005) [41]; Chenjiahe Group arc-type rocks from Hu (2005) [41]; South Qilian for-arc sediment from Tao et al. (2018) [42]; granite gneiss in Qinling Formation from Liu et al. (2013) [43]; granitoids in the Qilian basement from Huang et al. (2015) [37] and Xu (2007) [44].

Moreover, the Kangjiali gneissic granite has older Nd-isotopic model ages of 1.86–2.12 Ga, which is consistent with Proterozoic basement in Qilian Block, such as Longshan Group and Huluhe Group. Therefore, the petrogenesis of these volcanic rocks might be attributed to the addition of the ancient crustal materials in the Qilian Block, which might include the Precambrian Longshan Group and Huluhe Group. The granites have relatively high La/Nb

(3.63–4.64) and low La/Ba (0.09–0.21) ratios, indicating a probable source of subduction-modified continental lithospheric mantle [45].

Thus, the genetic source of this granitic rock is possibly the partial melting of a hybrid crustal source, which is mainly ancient meta-basement in North Qilian Orogen, mixing with relatively juvenile basaltic lower crustal materials formed by subduction of back-arc seafloor from Huluhe Group during the subduction period of the Shangdan oceanic crust towards the North China plate in the early Paleozoic time.

### 5.1.2. The Liwanxincun Metabasalt

The Liwanxincun metabasalt is characterized by medium- to low-K tholeiitic with LILEs enrichment and obvious negative Nb anomalies (Figures 6 and 10) such as the features of mantle-derived back-arc rock patterns [46], which indicate that they were mantle-derived in back-arc setting. The metabasalt shows arc-like patterns of LREE that are slightly enriched but are more affinitive to those E-MORB-like basalts with nearly flat REE patterns (Figure 9).

Concentrations of REE can be used to provide evidence for the presence of garnets in the magma source area [47]. Garnet preferentially incorporates HREEs over MREEs and LREEs. Consequently, magma derived from a garnet-bearing source should have high LREE/HREE and MREE/HREE ratios. In this study, $La_N/Yb_N$ ratios are in the ranges of 1.62–1.94 for Liwanxincun basalts, which shows that the residual garnets are insignificant. Moreover, the Dy/Yb ratios of lamprophyres and diorites are 1.85–1.98, which is between the spinel stability field (Dy/Yb > 2.5) and the garnet stability field (Dy/Yb < 1.5) [48]. Hence, garnet and spinel coexisted in the mantle source area of the Liwanxincun basalt.

As discussed above, the trace elements patterns of Liwanxincun basalt are similar to E-MORB (Figures 9 and 10). This is comparable with Qilian back-arc ridge in Nd isotope compositions of these basalts, further suggesting that their source might be MORB-like asthenospheric mantle. However, they also show arc-type features, including the depletion of Nb and enrichment of Rb, Ba, Pb and Sr [49,50], implying involvement of slab-derived components in magma generation. Therefore, their mantle source might be MORB-like asthenospheric mantle and might be influenced by slab-derived fluid. This is supported by transitional plots between MORB and OIB (Figure 16) and a slight enriched trend from MORB to CAB (Figure 17).

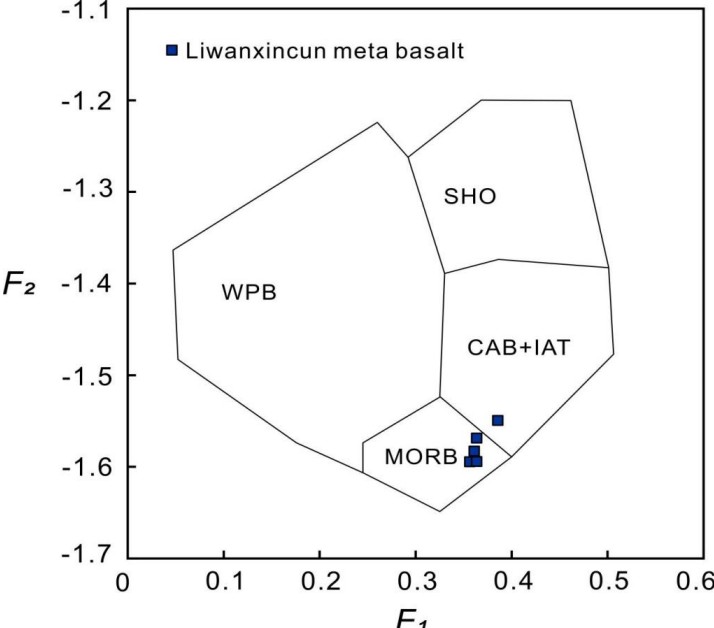

**Figure 16.** F1–F2 diagram [51] of Liwanxincun metabasalt.

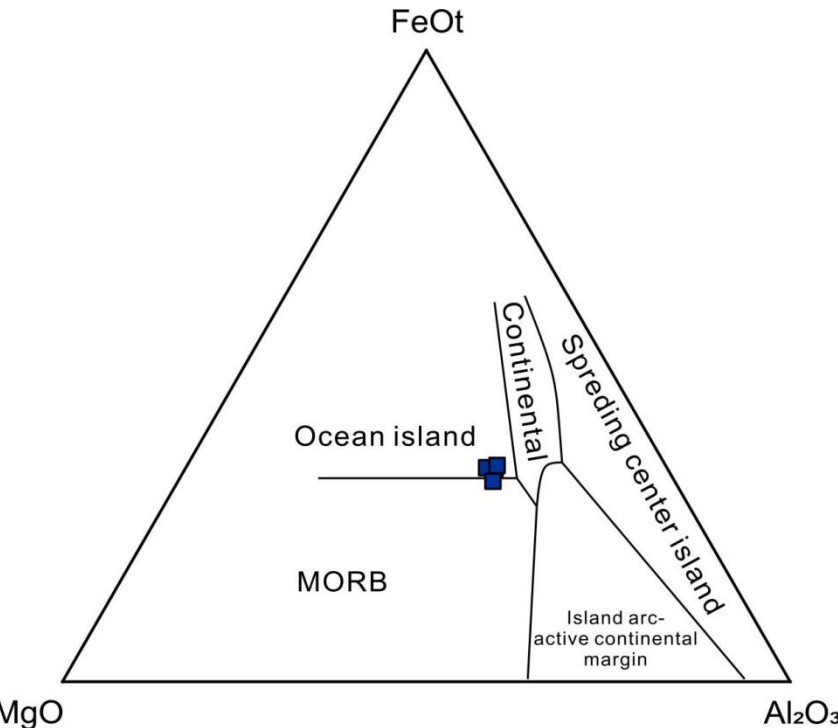

**Figure 17.** MgO-FeO$_t$-Al$_2$O$_3$ diagram [52] of Liwanxincun metabasalt.

Previous studies confirmed that LILEs such as Ba, Sr and Rb are preferentially incorporated into aqueous fluid [53,54], while Th, Ce and LREE are more efficiently transferred from the slab only when sediment melt is involved [54,55]. Liwanxincun metabasalt shows minimal enrichment in aqueous fluid mobile elements by high Ba/La and Ba/Th, low Th/Yb and Th/Nb, indicating the aqueous fluid involvement was significantly participated in and could be the main transport agency for slab-derived materials.

The intermediate between the N-MORB and the E-MORB trace element patterns of the Liwanxincun metabasalt could not be attributed to the overprinting of the slab-derived aqueous fluids. The flat and MORB-like REE patterns (Figure 9) also contradict significant crustal assimilation. For upwelling asthenospheric materials, it is widely believed that their interaction with the SCLM is a common process [56]. Thus, a relatively enriched source such as subcontinental lithospheric mantle (SCLM) might account for the enriched features of Liwanxincun metabasalt.

The scattered ($^{87}$Sr/$^{86}$Sr)$_i$ values toward MORB for Liwanxincun basalt (Figure 15) suggest a derivation of subduction sediments and contribution from the mixed mantle source, which includes a single derivation from depleted asthenospheric magma or lithospheric mantle-derived component. Therefore, the asthenospheric mantle-derived magma and slab-modified lithospheric mantle component were both involved in generation of Liwanxincun basalt.

In summary, Liwanxincun metabasalt might have been generated by ascended asthenosphere-derived magmas interacting with metasomatized SCLM, by which they imprinted signature of slab and MORB.

### 5.2. Tectonic Units in Sunjiaxia-Tianshui Area

#### 5.2.1. Yanjiadian-Xinjie Continental Arc

The existence of Yanjiadian-Xinjie continental arc is indicated by the exposure of large amounts of subduction-related diorites and granitoids intruded into the Longshan Group distributed along the northern side of the NQOB at the Sunjiaxia, Yanjiadian, Gongmen, Malu, Guanshan and Xinjie in the eastern margin of NQOB within the connection zone (Figure 18).

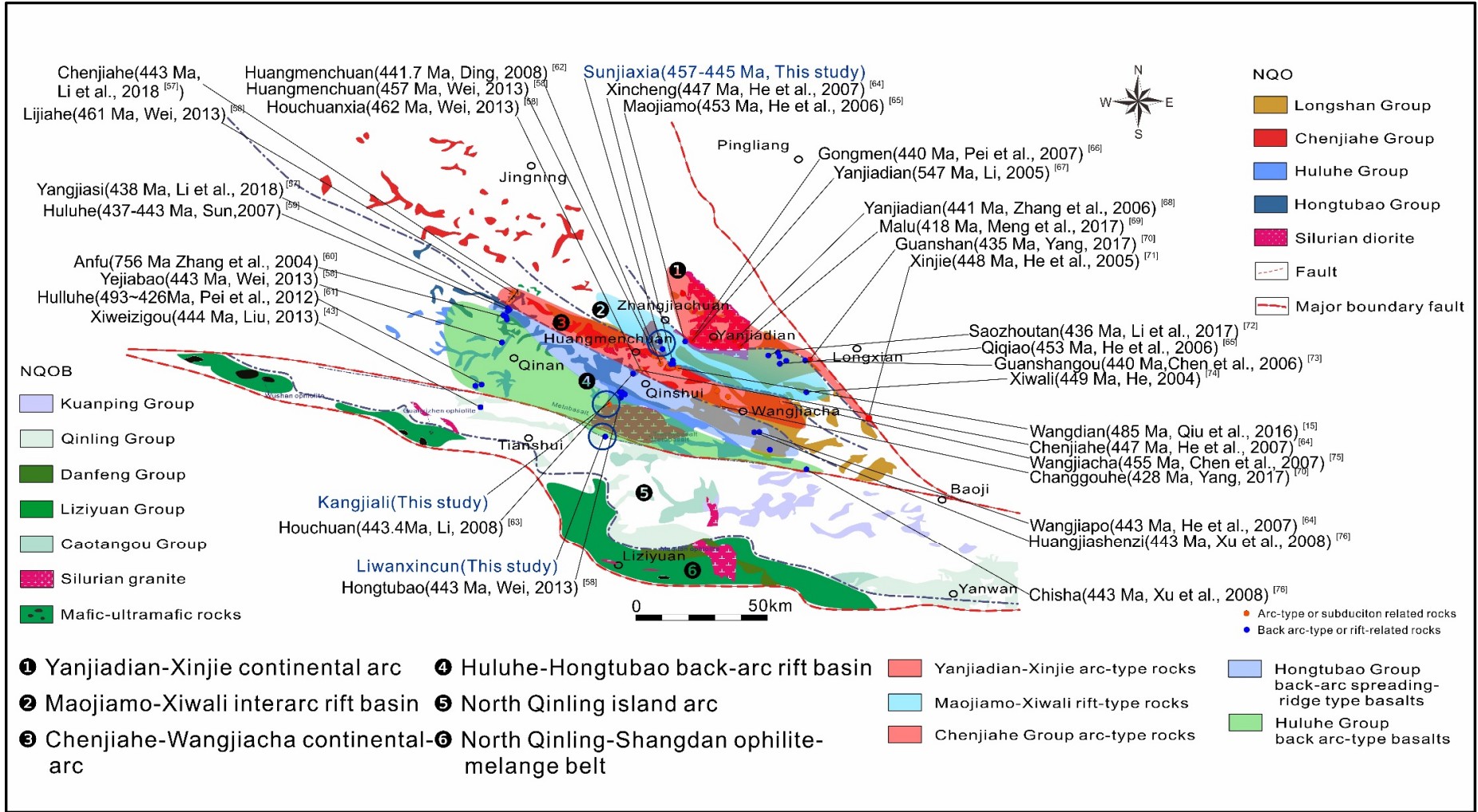

**Figure 18.** Tectonic setting map showing the ages and distributions of Early Paleozoic igneous rocks and the division of tectonic units in the Sunjiaxia-Tianshui area. The corresponding ages, methods, locations and references [7,15,43,57–76] are listed in Table 1.

**Table 1.** Summary of zircon U-Pb ages and tectonic implications of igneous rocks in the connection zone of NQO and WQOB.

| Pluton Name | Tectonic Unit | Position | Lithology | Dating Method | Age and Error (Ma) (Ma) | Author(s) | Interpretation |
|---|---|---|---|---|---|---|---|
| **Yanjiadian-Xinjie continental arc** | | | | | | | |
| Yanjiadian | North Qilian | Yanjiadian | Quartz diorite | Whole rock Rb-Sr | 547.7 ± 68.8 | Li, 2005 [67] | Continental arc |
| Shangping | North Qilian | Xinjie | Granitic gneiss | Zircon U-Pb (LA-ICP-MS) | 447 ± 5.4 | He, 2004 [74] | Continental arc |
| Xinjie | North Qilian | Xinjie, Zhangligou | Granite gneiss | Zircon U-Pb (LA-ICP-MS) | 447.7 ± 5.4 | He et al., 2005 [71] | Continental arc |
| Yanjiadian | North Qilian | Yanjiadian | Granite | Zircon U-Pb (SHRIMP) | 441 ± 10 | Zhang et al., 2006 [68] | Continental arc |
| Yanjiadian | North Qilian | Gongmen | Quartz diorite | Zircon U-Pb (LA-ICP-MS) | 440.2 ± 0.92 | Pei et al., 2007 [66] | Continental arc |
| Guanshan | North Qilian | Guanshan | Gneissic biotite granite | Zircon U-Pb (LA-ICP-MS) | 435 ± 2 | Yang, 2017 [70] | Subduction-related I-type granites |
| Yanjiadian | North Qilian | Malu | Diorite | Zircon U-Pb (LA-ICP-MS) | 418.1 ± 2 | Meng et al., 2017 [69] | High-Mg adakites |
| **Maojiamo-Xiwali interarc rift basin** | | | | | | | |
| Sunjiaxia | North Qilian | Sunjiaxia | Gneissic granite | Zircon U-Pb (LA-ICP-MS) | 457.0 ± 1.6, 445.9 ± 2.1 | This study | New age constraints of the rifting process |
| Qiqiao | North Qilian | Qiqiao | Plagioclase amphibolite | Zircon U-Pb (LA-ICP-MS) | 452.8 ± 1.7 | He et al., 2006 [65] | Extension-related |
| Maojiamo | North Qilian | Maojiamo | Plagioclase amphibolite | Zircon U-Pb (LA-ICP-MS) | 452.8 ± 1.7 | He et al., 2006 [65] | Extension-related |
| Xiwali | North Qilian | Tiefosi | Diabase dike | Zircon U-Pb (LA-ICP-MS) | 449.4 ± 5.4 | He, 2004 [74] | E-MORB |
| Xincheng | North Qilian | Xincheng | Basalt | Zircon U-Pb (SHRIMP) | 447 ± 8.5 | He et al., 2007 [64] | Back arc basin |
| Guanshangou | North Qilian | Guanshangou | Plagioclase amphibolite | Zircon U-Pb (LA-ICP-MS) | 440 ± 1.7 | Chen et al., 2006 [73] | Extension-related |
| Saozhoutan | North Qilian | Saozhoutan | Gabbro | Zircon U-Pb (LA-ICP-MS) | 436 ± 3.8 | Li et al., 2017 [72] | Back arc basin |
| Xincheng | North Qilian | Xincheng | Basalt | Zircon U-Pb (SHRIMP) | ~447–448 | Li, 2008 [63] | EM II and Rift-related |
| **Chenjiahe-Wangjiacha continental arc** | | | | | | | |
| Wangdian | | Wangdian | Tonalite | Zircon U-Pb (LA-ICP-MS) | 485.3 ± 6.2 | Qiu et al., 2016 [15] | Continental arc |
| Houchuanxia | | Houchuanxia | Dacite | Zircon U-Pb (LA-ICP-MS) | 462.4 ± 3.4 | Wei, 2013 [58] | Continental arc |
| Lijiahe | | Lijiahe | Dacite | Zircon U-Pb (LA-ICP-MS) | 461.2 ± 3.1 | Wei, 2013 [58] | Continental arc |
| Huangmenchuan | | Huangmenchuan | Granodiorite | Zircon U-Pb (LA-ICP-MS) | 457.0 ± 3.2 | Wei, 2013 [58] | Continental arc |
| Wangjiacha | | Dianzishang | Quartz diorite | Zircon U-Pb (LA-ICP-MS) | 454.7 ± 1.7 | Chen et al., 2007 [75] | Subduction-related |
| Xincheng | | Xincheng | Dacite | Zircon U-Pb (SHRIMP) | 447 ± 8.5 | He et al., 2007 [64] | Continental arc |
| Chenjiahe | | Chenjiahe | Dacite | Zircon U-Pb (SHRIMP) | 443 ± 3.8 | Li et al., 2018 [57] | Continental arc |
| Huangmenchuan | | Huangmenchuan | Biotite monzogranite | Zircon U-Pb (LA-ICP-MS) | 441.7 ± 3.4 | Ding, 2008 [62] | Continental arc |
| Huangmenchuan | | Huangmenchuan | Granodiorite | Zircon U-Pb (LA-ICP-MS) | 440.5 ± 4.4 | Wei, 2013 [58] | Continental arc |
| Changgouhe | | Changgouhe | Quartz monzonite | Zircon U-Pb (LA-ICP-MS) | 428 ± 2 | Yang, 2017 [70] | Subduction-related I-type granites |
| **Huluhe-Hongtubao back-arc rift** | | | | | | | |
| Yangjiasi | North Qilian | Yangjiasi | Basalt | Zircon U-Pb (SHRIMP) | 438 ± 7 | Li et al., 2018 [57] | Back-arc basin |
| Wangjiapo | North Qilian | Wangjiapo | Basalt | Zircon U-Pb (LA-ICP-MS) | 443.4 ± 1.7 | He et al., 2007 [64] | Initial back-arc basin |
| Houchuan | North Qilian | Hongtubao | Basalt | Zircon U-Pb (LA-ICP-MS) | 443.4 | Li, 2008 [63] | Back-arc basin |
| Huangjiashenzi | North Qilian | Huangjiashenzi | Basalt | Zircon U-Pb (LA-ICP-MS) | 443.4 ± 1.7 | Xu et al., 2008 [76] | Back-arc basin |
| Huluhe | North Qilian | Yangjiasi | Metabasalt | Zircon U-Pb (LA-ICP-MS) | 436.9 ± 5.7, 443.4 ± 1.7 | Sun, 2007 [59] | Back-arc basin |
| Caochuanpu | North Qilian | Caochuanpu | Monzogranite | Zircon U-Pb (SHRIMP) | 434 ± 10 | Wei, 2013 [58] | Subduction-collision-related A-type granite |

**Table 1.** *Cont.*

| Pluton Name | Tectonic Unit | Position | Lithology | Dating Method | Age and Error (Ma) (Ma) | Author(s) | Interpretation |
|---|---|---|---|---|---|---|---|
| Kangjiali | North Qilian | Kangjiali | Gneissic granite | | | This study | New geochemical constraints on the evolution process of the back-arc basin with subduction-related I-type granite features |
| **Huluhe-Hongtubao back-arc basin** | | | | | | | |
| Huluhe | West Qinling | Huluhe | Gabbro | Whole rock Rb-Sr | 756 ± 12 | Zhang et al., 2004 [60] | Plume-type back arc basin ophiolite |
| Xiweizigou | West Qinling | Xinyang | Metagabbro | Zircon U-Pb (LA-ICP-MS) | 443.6 ± 1.8 | Liu, 2013 [43] | SSZ-type ophiolite |
| Chisha | West Qinling | Chisha | Metabasalt | Zircon U-Pb (LA-ICP-MS) | 443.4 ± 1.7 | Xu et al., 2008 [76] | Back-arc basin |
| Huluhe | West Qinling | Yejiabao | Metabasalt | Zircon U-Pb (LA-ICP-MS) | 443.4 ± 1.7 | Wei, 2013 [58] | Back-arc basin |
| Huluhe | West Qinling | Huluhe | Metamorphosed quartz sandstone | Zircon U-Pb (LA-ICP-MS) | 426–493 (subduction-collision period) | Pei et al., 2012 [61] | Subduction-collision-related |
| Liwanxincun | West Qinling | Liwanxincun | Metabasalt | | | This study | New geochemical constraints on the subduction process of North Qinling-Shangdan Ocean with Plume-type back-arc basin ophiolite |
| **North Qinling island arc** | | | | | | | |
| Caotangou G. | West Qinling | Sangyuan | Intermediate-basic volcanics | Zircon U-Pb (LA-ICP-MS) | 456 ± 2 | Wang et al., 2007 [77] | Subduction-related |
| Huangniupu | West Qinling | Honghuapu | Quartz monzonite | Zircon U-Pb (LA-ICP-MS) | 440 ± 3 | Yao et al., 2017 [78] | Subduction-related I-type granites |
| Yangjiazhuang | West Qinling | Yangjiazhuang | Quartz diorite | Zircon U-Pb (LA-ICP-MS) | 439 ± 3 | Ren et al., 2018 [79] | Subduction-related high Ba-Sr diorites |
| Honghuapu | West Qinling | Honghuapu | Quartz diorite | Zircon U-Pb (LA-ICP-MS) | 438 ± 3 | Ren et al., 2018 [79] | Subduction-related high Ba-Sr diorites |
| Baihua | West Qinling | Baihua | Gabbro | Zircon U-Pb (LA-ICP-MS) | 435 ± 2 | Pei et al., 2007 [80] | Subduction-related |
| **North Qinling-Shangdan melange belt** | | | | | | | |
| Yanwan | West Qinling | Yanwan | Gabbro | Zircon U-Pb (LA-ICP-MS) | 518 ± 3 | Dong et al., 2011 [14] | E-MORB-type ophiolite |
| Guanzizhen | West Qinling | Guanzizhen | Gabbro | Zircon U-Pb (LA-ICP-MS) | 512 ± 2 | Yang et al., 2018 [7] | Plume-type ophiolite |
| Guanzizhen | West Qinling | Guanzizhen | Metagabbro | Zircon U-Pb (SHRIMP) | 489 ± 10 | Li, 2008 [63] | Subduction-related |
| Wushan | West Qinling | Hualingou | Gabbro | Zircon U-Pb (SHRIMP) | 440 ± 5 | Li, 2008 [63] | Subduction-related |

These subduction-related volcanic rocks are mostly characterized by geochemical compositions with depletion of HFSE (Nb, Ta, Zr and Hf) and enrichment of LREE, represented by the Yanjiadian diorites (441 Ma [68]; 547.7 Ma [81]), Malu Diorite (418.1 Ma [69]) and Xinjie granitic gneiss (447 Ma [74]).

The Yanjiadian diorites have geochemical characteristics of island-arc-type rocks and formed in continental arc setting [68]. Qiu (2016) [15] suggests that the incorporating melts of Wangdian tonalite from the mantle wedge and occur in continental margin. He (2004) [74] reported that the Xinjie gneiss is characterized by enrichments in LREE and LILEs and depletions in HREE, showing that they were generated by subduction-related magmatism in an arc setting. The subduction-related Xinjie gneiss yields a zircon U-Pb age of 447 ± 5.4 Ma [71] (Figure 18, Table 1).

In any case, these subduction-related rocks share a similar chronology and petrogenesis, suggesting that they originated within the same tectonic unit as the continental arc belt. The ages of these rocks are within the subduction period of North Qilian-Kuanpin Ocean from ~520 Ma [82] to ~495–443 Ma [83–86]. Notably, all these island-arc-type rocks are confined to the northern side of the Chenjiahe island arc belt in lateral discussion. Thus, in the north of Chenjiahe continental arc, Yanjiadian-Xinjie continental arc belt might possibly be generated by the southward subduction of the North Qilian Ocean within 547.7–418 Ma.

5.2.2. Maojiamo-Xiwali Interarc Rift Basin

To the south of Yanjiadian-Xinjie continental arc belt, a large number of extension-related mafic rocks and granitoids is typically exposed in the southern margin of Yanjiadian-Xinjie continental arc belt from west to east, which are the Maojiamo plagioclase amphibolite [65] and Xincheng basalt [63,64] in Gongmen area, Guanshangou plagioclase amphibolite [73], Saozhoutan gabbro [72,73], Qiqiao plagioclase amphibolite [65] in Guanshangou area and Xiwali diabase dike [74] in Xinjie area, etc. (Figure 18, Table 1).

Previous petrological and geochemical studies reported that these mafic rocks show a similar elemental and isotopic geochemistry signature within plate basalts, such as enrichments in LREE and LILEs (K, Rb, Ba, Th, etc.), depletions in HF, Zr, Y and Yb, and low total amount of rare earth elements ($\Sigma$REE less than $100 \times 10^{-6}$), and it is proposed that these rocks were formed in a back-arc basin [73]. They show intermediate geochemical features between MORB and IAB, indicating that they were formed in a back-arc basin setting [72]. Additionally, a couple of gneissic granites (Sunjiaxia) (Table 1) and rhyolites from Chenjiahe Group outcrop in this belt. These rhyolites with arc-type affinity are identified with enrichment of LREE, LILE (Ru, Ba, Th, U and K) and depletion of HFSE (Nb, Ta, Ti) [64].

In view of the co-occurring of the above basalts and granites but lack of andesite that is normally outcropped in the island arc belt, the volcanic assemblage in Xincheng area can be inferred to be bimodal volcanics and is derived from rift setting within an island arc [60]. This is supported by the basalts with mixed source signatures between depleted mantle and enriched mantle from Chenjiahe Group in Xincheng area [64], which are interpreted as products of inter-arc rifting.

These petrological and geochemical features indicate they were formed either in a rift setting [63,64] or in a transitional setting where an initial continent rift was expanding into a mature oceanic basin [73]. U-Pb zircon ages of 452.8 ± 1.7 Ma [65], 436 ± 3.8 Ma [72], 449.4 ± 5.4 Ma [74], 442.8 ± 1.1 Ma and 440 ± 1.7 Ma [73], as well as Zircon U-Pb (SHRIMP) age of 447 ± 8.5 Ma [64] give limitations to the rifting-related mafic rocks. For the acid rocks, the crystallization ages of the Sunjiaxia gneissic granite are estimated as 457.0 ± 1.6 Ma and 445.9 ± 2.1 Ma. This age is in accordance with the age of dacites in Xincheng estimated as 448 ± 8 Ma [63].

All these age data are within the timing of the Yanjiadian-Xinjie subduction-related volcanics discussed above, which shows rifting in respond to the subduction during 457–436 Ma (Table 1). Based on the petrological and geochemical studies above, these rock

associations might record the interarc rifting process. Thus, the Maojiamo-Xiwali interarc rift basin was generated along the southern margin of Yanjiadian-Xinjie continental arc belt.

### 5.2.3. Chenjiahe-Wangjiacha Continental Arc

In the Chenjiahe-Wangjiacha area, detailed studies indicate that a typical island-arc volcanic zone exists. Several dioritic-dacitic intrusions with typical island-arc geochemical signatures, known as Chenjiahe Group, are distributed along the belt, from west to east, at Lijiahe [58], Chenjiahe [41,57], Huangmenchuan [62], Wangdian [15], Xincheng [63,64], Houchuanxia [58], Changgouhe [70] and Wangjiacha [75] (Figure 18).

Both diorites and dacites from this belt display enrichment of LREE and LILEs and high-degree fractionation of HFSE with negative Nb–Ta, Sr and Ti anomalies, as well as slight enrichment of Sr-Nd-Hf isotopic compositions [41,58,63]. These indicate that they were generated by partial melting and binary mixing of ancient crustal and juvenile materials in a continental arc setting [57]. Detailed geochemical studies revealed that these mafic intrusions were derived from a mantle wedge source above the subducted slab [60] and experienced mixing with crustal materials [57,75], which show genetic connection with Yanjiadian-Xinjie continental arc. This is also supported by the geochemical characteristics of enrichment in LREE and LILEs, significant depletion of Nb, Ta and Ti, and enrichment of Th, Pb, Sr and Sr-Nd isotopic compositions in the intermediate to felsic rocks from Wangdian [15], Changgouhe [70] and Wangjiacha [75] arc-related intrusions, respectively. Therefore, it is proposed that the dioritic-dacitic rocks were formed in a continental arc.

Other than a U-Pb age of 485.3 $\pm$ 6.2 Ma [15] from Wangdian, the other zircon U-Pb ages of arc-type volcanic rocks from Chenjiahe Group ranging from 458 to 428 Ma were from Heihe area (Table 1). The subduction-related plutons from Chenjiahe Group yield zircon U-Pb ages ranging from 454.7 to 428 Ma and zircon U-Pb (SHRIMP) age of 447 $\pm$ 8 Ma (Table 1). These ages suggest the subduction mainly occurred within 458–428 Ma.

The geochemical characteristics and geochronical studies of Chenjiahe and Wangjiacha arc-related rocks reveal an eastward extension of Chenjiahe continental arc belt parallel with Yanjiadian-Xinjie continental arc belt during 458–428 Ma.

### 5.2.4. Huluhe and Hongtubao Back Arc Rift Basin

Huluhe-Hongtubao back-arc basin is distributed along the south margin of Chenjiahe-Wangjiacha continental arc belt and mainly comprises basalts from Hongtubao Formation and metabasalts from Huluhe Formation (Figure 18).

The Hongtubao basalts are characterized by LREE and LILE enrichments and Nb-Ta depletions, suggesting sediments had been carried into the mantle during subduction [58]. The tholeiitic basalts of Hongtubao Formation, represented by basalts at Lijiahe-Yangjiasi area [57,58], Huangmenchuan-Xincheng area [58,63] and Qingshui [64] area, display medium- to low-K tholeiitic, LREE and LILE enrichments and Nb-Ta depletions and MORB-like Sr-Nd-Hf isotopic compositions. Therefore, they are inferred to be derived from a depleted mantle in a spreading center but imposed by subducted material in an initial back-arc setting [41,57,64]. In addition, studies revealed that pillow basalts at Hongtubao, characterized by both N-MORB affinity and subduction-related geochemical features such as medium- to low-K, LREE and LILE enrichments and negative Nb, Ta and Ti anomalies, are derived from a depleted mantle imposed by subduction-related fluids in a developed back-arc basin [58].

The basalts in the Xincheng yield a zircon U-Pb SHRIMP age of 447 $\pm$ 8 Ma [63], which is in accordance with an ICP/MS U-Pb zircon age of 443.4.5 $\pm$ 1.7 Ma from basalts in Wangjiapo in the eastern reach of this belt [64]. Based on the SHRIMP U-Pb zircon age, the gabbro and MORB-type plagiogranite of Hongtubao give ages of 500 $\pm$ 3 Ma and 443.4 $\pm$ 1.7 Ma, respectively [58,87]. The age of Xincheng basalts is slightly older than the formation age of Hongtubao basalt, indicating the Hongtubao back-arc rifting is in response to the subduction and is genetically related to the development of Chenjiahe-Wangjiacha arc. In the Xiweizigou-Moshigou area, the metabasalts have been dated by

ICP/MS, which gave zircon U-Pb ages of 443.6 ± 1.8 Ma and 432.6 ± 3.8 Ma [43]. They are characterized by typical E-MORB features such as enrichments in LREE and LILEs, mild depletions in Nb and flat HFSE patterns, which were interpreted to be formed in the late stage of a back-arc basin [88]. The ophiolites from Huluhe Group can be generated in Neoproterozoic-Cambrian time, which is reported by detrital zircon age spectrum of 801–445 Ma in Huluhe area [89].

Together with the above ages of rocks from back-arc basin setting, the formation timing of the Hongtubao back-arc basin was from the latest Neoproterozoic to ca. 433 Ma, which shows consistency with the arc-like volcanic rocks with the North Qinling arc belt. The similar age period between the two belts shows they are closely related to each other and are both in the arc-back arc system of North Qinling Orogen in the subduction background of the northward subduction of North Qinling-Shangdan Ocean during Early to Middle Paleozoic time.

Considering all above data, we infer that the formation timing of the back-arc spreading ridge can be constrained between 447 Ma and 438 Ma, which shows consistency with the back-arc basin metabasalts within Huluhe mature back-arc basin developed from latest Neoproterozoic to ca. 433 Ma. The similar age period between the two belts shows they are closely related to each other and are both in the back-arc basin in the subduction background.

### 5.2.5. North Qinling Island Arc

The North Qinling island arc belt is located between the North Qinling-Shangdan suture zone and is recognized by early Paleozoic subduction-related volcano-sedimentary sequences, igneous complexes and intermediate-felsic plutons, which intruded in Paleoproterozoic metamorphic basement of the Qinling Group [90].

The North Qinling island arc belt is characterized by arc-type igneous complexes, such as the Baihua gabbros with LA-ICPMS U-Pb zircon ages of 450–435 Ma and Liushuigou gabbros with U-Pb zircon ages of 549–508 Ma, as well as a TIMS zircon U-Pb age of 508 ± 3 Ma [91]. Meanwhile, Honghuapu (414 Ma) [15] and Huangniupu (440 ± 3 Ma [92]) I-type granites [78], as well as Yangjiazhuang (439 Ma) and Honghuapu (438 Ma) high-Ba-Sr diorites [79] give further time constraint of this island arc belt.

The diorites and gabbros from Baihua show enrichment of LREE and LILEs, and high-degree fractionation of HFSE with depletion of Nb and Ti [93]. Together with co-occurring basalts from the Caotangou Group with enriched LREE and LILEs and significantly depleted Nb, Ta, Zr and Hf [77,94], this volcanic belt is indicated to have been formed in an island arc [79,93]. Additionally, subduction-related intrusions are documented as being derived from a mantle wedge source above subducted slab, which is represented by the Guanzizhen [94].

These complexes have zircon U-Pb ages of range from 549 Ma to 435 Ma (Table 1), which shows the subduction period to at least start from the Cambrian and last to the late Ordovician. All the ages and geochemical signatures of gabbros, andesites and diorites above indicate that the North Qinling island arc belt was generated by northward subduction of the North Qinling-Shangdan Ocean beneath the North Qinling terrain (NQT) during ca. 549 Ma to ca. 435 Ma.

### 5.2.6. North Qinling-Shangdan Ophiolitic Melange Belt

The North Qinling-Shangdan Ophiolitic melange belt is located at the southern margin of the Western Qinling Orogen and is marked by a couple of discontinuously exposed ophiolitic assemblages and metamorphosed mafic and ultramafic arc-related volcanic rocks named as the Liziyuan Group (Figure 18).

The investigations reveal that the ophiolite sequence consists mainly of metamorphosed mafic and ultramafic rocks outcropping, from west to east, at Wushan, Guanzizhen, Muqitan and Yanwan area. The Guanzizhen ophiolite exhibits typical characteristics of the N-MORB, with flat REE patterns and slight depletion of LREE, which is interpreted to be

formed in a mid-ocean ridge setting [18]. The Muqitan, Wushan and Yanwan ophiolites show E-MORB geochemical features of slightly enriched LREE and flat HREE and HFSE patterns and were interpreted to be formed in a spreading oceanic lithosphere setting, including an initial oceanic spreading center of a mid-oceanic ridge (Yanwan) [14] and transitional ridge segments of a mid-ocean ridge (Wushan) [14].

The timing of the North Qinling-Shangdan Ophiolitic melange has been constrained by many studies, ranging from 763 Ma to 440 Ma, recording the seafloor spreading of this ocean. The Guanzizhen ophiolite yields LA-ICPMS U-Pb zircon ages of 512–471 Ma [7,95]. The Muqitan, Wushan and Yanwan ophiolites yield LA-ICPMS U-Pb zircon ages of 763–440 Ma [14,63].

These geochronological data suggest that the initial formation of North Qinling-Shangdan Ocean could be at ca. 763 Ma. This oceanic crust evolved into a mature N-MORB setting by ca. 534–471 Ma, and at least existed until 440 Ma.

### 5.3. Tectonic Evolution in Sunjiaxia-Tianshui Area

Based on the tectonic unit division discussed above and the field relationships, a model is proposed for the Paleozoic tectonic evolution of the connection zone between North Qilian Orogen (NQO) and West Qinling Orogen (WQO) (Figure 19).

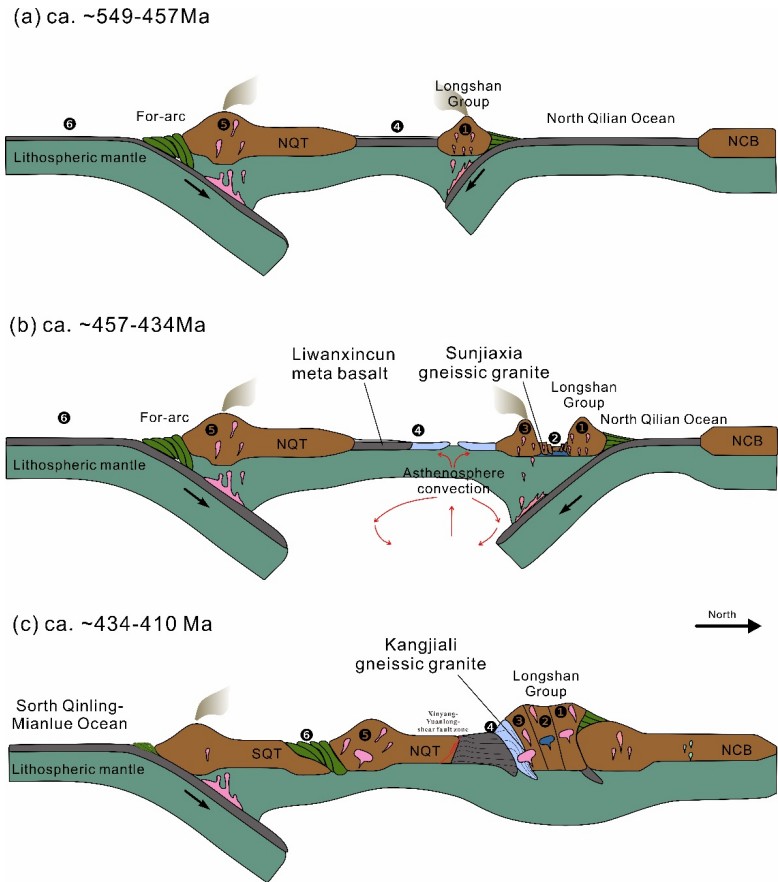

**Figure 19.** A tectonic model for Early Paleozoic (~549–410 Ma) development of the Sunjiaxia-Tianshui area, involving generation of continental island arc and back-arc basin, closure of oceanic plate and back-arc basin and collision between the NCB and NQT. (**a**) The ca. ~549–457 Ma Ma subduction of the North Qilian oceanic plate and the generation of the Yanjiadian-Xinjie continental arc; (**b**) the ca. ~457–434 Ma generation of the Maojiamo-Xiwali inter-arc rift basin and coeval Chenjiahe-Wangjiacha continental island arc, followed with the spreading of Hongtubao back-arc basin ridge; (**c**) the ca. ~434–410 Ma closure of the Huluhe-Hongtubao back-arc basin and collision between the NQT and NCB.

5.3.1. Opening, Spreading and Subducting of North Qilian Ocean, North Qinling Ocean and Huluhe Back-Arc Basin from Neoproterozoic to Late Ordovician

Opening and Subducting of the North Qilian Ocean from Neoproterozoic—800 Ma to 476 Ma

The NQO is an oceanic suture zone between the Qilian Block and the North China Craton and is suggested to have been part of the Proto-Tethys Ocean before 497 Ma [96].

Previous studies have revealed that the North Qilian Ocean may have initially started its expansion from small ocean basin since the Meso-Proterozoic, represented by the N-MORB-type gabbro from Jiugequan ophiolites [97]. As represented by the arc magmatism of 517–490 Ma in North Qilian Orogen [3], the ocean basin of North Qilian Ocean might have been opened during the late Neoproterozoic to Early Cambrian.

NQO possibly started to subduct towards the south during 547–476 Ma, which is indicated by a series of subduction-related institutions separated in the northern part of North Qilian orogenic belt including Yanjiadian diorite (ca. 547 Ma) [67], Kekeli plagioclase granite (ca. 512 Ma) and quartz diorite (ca. 501–476 Ma) [27], Yemazui granite (ca. 508 Ma) [27] and Niuxinshan granite (ca. 477 Ma) [83].

Previous studies have revealed that the Qilian Block had formed by rifting from the East Gondwana during the Neoproterozoic and experienced transformation in tectonic setting from an arc environment on an active continental margin (ca. 900 Ma) to a continental rift environment (ca. 800 Ma) [98]. As represented by the 534 Ma oceanic gabbros from the Guanzigou area, the Shangdan Ocean was already established before 534 Ma, and the ocean separated the South China block from the North China block.

Opening and Subducting of the North Qinling Ocean from 763 Ma to 455 Ma

The North Qinling-Shangdan ophiolitic melange is located between the South Qinling Terrane (SQT) and NQT and represents the remnants of oceanic crust of Shangdan ocean [14], which could be a branch of the Proto Tethyan Ocean separating the South China from the North China blocks.

Some Cambrian–Ordovician ophiolites were reported from the Guanzizhen (ca. 512 Ma) [7] and Wushan (ca. 456 Ma) [63]. Accordingly, the spreading of the Shangdan Ocean might have occurred in Cambrian time and it might have still been expanding in Ordovician times.

During the spreading process of North Qinling Ocean, the NQT turned into an island volcanic arc (ca. 549 Ma [91]) and was then accreted onto the NCB. The oceanic lithosphere of North Qinling-Shangdan Ocean was still subducting towards the north beneath NQT at ca. 508 Ma, revealed by Liushuigou intermediate-basic meta-igneous rocks in Guanzizhen area [94]. The timing of subduction in early Paleozoic can be constrained by Guanzizhen subduction-related gabbro and Miaogou back-arc basin type pyroxenite, which yields TIMS zircon U-Pb ages of 507.5 ± 3.0 Ma [94], and LA-ICPMS zircon ages of 485 ± 5 Ma [99], respectively.

Opening and Spreading of Huluhe Back Arc Basin from 756 Ma to 443.6 Ma

In Meso-Neoproterozoic time, the Huluhe back-arc basin evolved into a small-scale oceanic basin, generated the plume-type back arc basin ophiolite with whole rock Rb-Sr age of 756 ± 12 Ma in Huluhe area [60]. This back-arc basin separated the North Qinling Terrane from Longshan Group as the northward subduction of North Qinling-Shangdan Ocean.

From 493 to 443 Ma, there was still formation of oceanic crust in Huluhe area and this produced the back-arc type sediments, represented by the LA-ICP-MS detrital zircon U-Pb ages of 426–493 Ma from the metamorphosed quartz sandstone in Huluhe area [61].

5.3.2. Formation of Maojiamo-Xiwali Interarc Basin and Huluhe-Hongtubao Back-Arc Basin during the Early Paleozoic Time and Its Association with Connection Zone from 457 to 434 Ma

Tectonic Setting for the Formation of Continental Arc and Back-Arc Basin in Late Devonian

During the Early Paleozoic, the North Qilian Ocean between the NQO and NQOB could have been part of the Proto-Tethys Ocean [100]. The connection zone between

NQO and NQOB has been interpreted to be an Early Paleozoic oceanic suture zone [97] (Figure 20), which could be the northernmost orogenic collages of the Proto-Tethys Ocean. From the late Devonian to early Silurian (ca. 440 Ma), a series of micro-continental blocks, including the North China, Qilian, Alxa, North Qinling and Qaidam blocks in the eastern reach of Proto-Tethys Ocean, continually subducted southward and accreted to the northern margin of the Gondwana continent, which is in the context of the closure of the Proto-Tethys Ocean [57].

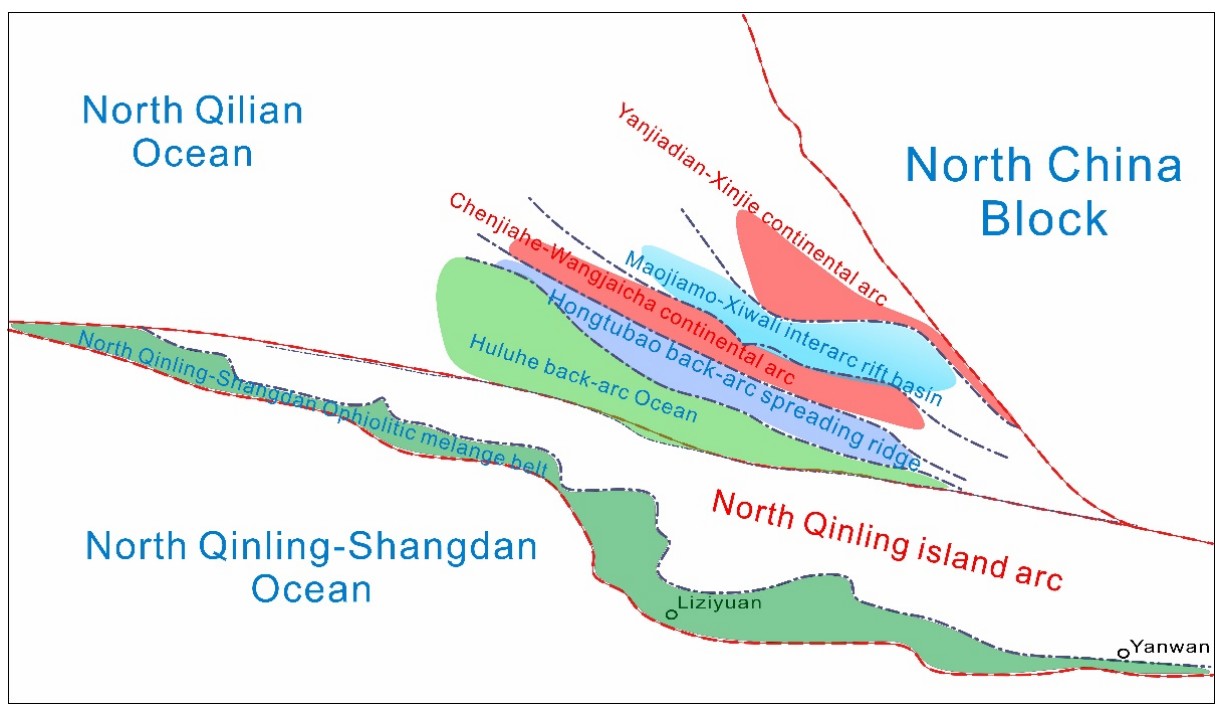

**Figure 20.** Paleogeographic map of the connection zone in Ordovican-Devonian time.

Represented by the 487 ± 9 Ma Dachadaban tholeiites [101] and the 447–430 Ma subduction-related complex from the Laohushan area [102], the North Qilian ocean was still spreading and subducting during ca. 458–454 Ma, and the ocean corresponds to the North Qinling tectonic unit and might link the NCB and Longshan Group, which have been identified as basement of the Qilian Block or NCB from their metamorphic ages, trace element features and detrital zircon spectrum [57,68].

To the south, the Huluhe Group, extending from Huluhe to Chisha between North Qinling Terrane and Longshan Group, is proposed to represent a back-arc oceanic basin [58,76]. The basalts from Xiweizigou (444 Ma [43]) and Huluhe (434 Ma [62]) appear to record a similar time span, from continual seafloor spreading of North Qinling-Shangdan Ocean (485 Ma) [99] to subduction in early Silurian (ca. 456–440 Ma) [63].

The development of the Huluhe-Hongtubao back-arc basin could be attributed to the spreading and subducting of the North Qinling-Shangdan Ocean in the southernmost of the connection zone; according to the geochemistry and geochronology, a series of early Silurian arc-type volcanic intrusions outcropped in Baihua (434.6 ± 1.5 Ma [93]), Honghuapu (438 Ma [79]) and Yangjiazhuang (439 Ma [79]).

Formation of Continental Arc and Back-Arc Basin

Before ca. 463 Ma, the North Qilian Ocean kept subducting as before, suggested by the formation of the Minleyaogou arc-type granodiorite in ca. 463 Ma [27] and a couple of subduction-related intrusions in Longshan Group discussed above.

As a result, on the northern edge of the Longshan Group, a subduction zone named Yanjiadian-Xinjie towards the south possibly existed and developed more arc-type intru-

sions such as Yanjiadian diorites (547.7–440.2 Ma, Table 1). The development of this arc belt caused extension and formation of the Maojiamo-Xiwali inter-arc rift basin discussed above (Figure 18).

To the south of the Yanjiadian-Xinjie area, detailed work indicates that metabasalts spreading from Maojiamo to Xiwali related to the depressional rift basin setting, which formed from ca. 457 to ca. 436 Ma (Table 1). This age span is in accordance with zircon U-Pb age of Sunjiaxia gneissic granites (457–446 Ma, this study), which formed near Maojiamo rift type plagioclase amphibolite within this rift basin. Following subduction, the outward migration of the arc location and the transition from an arc into a back-arc basin setting in a subduction zone over time are common [103].

To the south of Maojiamo-Xiwali interarc rift basin, the predominant subduction along the northern side of the Longshan Group might have brought migration of arc-type dioritic intrusions in Chenjiahe-Wangjiacha area at 514–508 Ma (Figure 18, Table 1), and might have resulted in the initial spreading of the Hongtubao back-arc ridge in the northern edge of Huluhe back-arc basin at 443–434 Ma (Table 1). Some of these arc-type intrusions also show genesis which is associated with the reworking of the Precambrian basement represented by the Longshan Group [57]. As the back-arc ridge spread between the Chenjiahe-Xiwali continental arc belt and North Qinling island arc belt, the Huluhe-Hongtubao back-arc basin came to exist (Figure 18).

At the same time, to the southern margin of connection zone, the further development of North Qinling island arc brought more volcanics and intrusions with oceanic island-arc geochemical signature, which is indicated by Caotangou intermediate-basic volcanics (456 Ma, Table 1) and Baihua gabbros (450 Ma, Table 1) exposed within the North Qinling-Shangdan ophiolitic melange belt. The northward subduction of North Qinling-Shangdan Ocean might have caused the seafloor of Huluhe-Hongtubao back-arc basin to be subducted northward beneath the Longshan Group and generate peraluminous arc-type Kangjiali gneissic granite. Represented by the oldest age of flysch formation of clastic rocks in the Huluhe Group correlated to the arc-continent collision (ca. 443 Ma [61]), the subduction reached a peak stage in the Silurian. The collision-related granitoids in this region such as the Caochuanpu granites (434 Ma [68]) might further constrain the ending of arc-continent collision.

### 5.3.3. Closure of the Oceanic Plates and Back-Arc Basin in Connection Zone from 434 Ma to 410 Ma

Until ca. 430 Ma, a series of collisions had resulted from the successive closure of the Erlangping back-arc basin and the Shangdan Ocean [21], which can be compared to Huluhe-Hongtubao back-arc basin.

In the connection zone, the extinction of the Huluhe-Hongtubao oceanic crust occurred at ca. 424 Ma, demonstrated by the youngest age group of detrital zircons of Huluhe Group (426 Ma [61]) and collision-related metamorphic rocks from the Qinling Complex (433−424 Ma [104]).

During the closure of the Huluhe-Hongtubao back-arc basin, the collision between the North China Block (NCB) and West Qinling Terrane (WQT) occurred in Mid-Silurian times along the northern margin of WQT. The collision is supported by the emplacement of granitic intrusions at 435–427 Ma (Table 1). Subsequent post-orogenic extension had lasted to ca. 422 Ma, represented by syn-collisional magmatic rocks from 460 Ma to 422 Ma [105] in the junction region of the North Qilian and Qinling orogenic belts.

All these tectonic units in the connection zone of NQT and WQO characterized the tectonic evolution of multiple oceanic plates or basins in the context of the convergence of NCB and southern micro-continental blocks such as Longshan Group and NQT during late Devonian-early Silurian times.

## 6. Conclusions

1.  The magma emplacement ages for the Sunjiaxia (068, 069) gneissic granite in the Maojiamo–Sunjiaxia area, northern part of the connection zone are $457.0 \pm 1.6$ and $445.9 \pm 2.1$ Ma, respectively. The ages of 457–445.9 Ma represent the timing of interarc rift basin in Maojiamo-Xiwali area.

2.  In the southern part of the connection zone, the Kangjiali gneissic granites have geochemical characteristics of island-arc-type rocks and were generated by melts from older crustal materials from Longshan Group with addition of a juvenile basaltic source from the lower crust which incorporated subduction-related fluids during the collisional process between Huluhe back-arc ocean floor and Northern Longshan Group basement. The Liwanxincun metabasalts exhibit E-MORB geochemical features and their formation resulted from magmas derived from partial melting of shallow asthenospheric mantle that interacted with slab fluid metasomatized mantle in the back-arc extension setting.

3.  We suggest that the connection zone can be divided into six tectonic units during the early Paleozoic time. The Yanjiadian-Xinjie continental arc represents the early period of subduction of North Qilian Ocean and the Maojiamo-Xiwali interarc rift is the product of the extension triggered by southward subduction in ca. 457–446 Ma, which afterwards led to the formation of Chenjiahe-Wangjiacha continental arc, as well as the Hongtubao back-arc rift in Huluhe back-arc basin. The tectonic evolution of the connection zone is closely associated with the closure of the North Qilian Ocean and North Qinling-Shangdan Ocean in the context of the convergence process of micro continental blocks, including North China Block, Longshan Group and North Qinling Terrane.

**Supplementary Materials:** The following supporting information can be downloaded at: https://www.mdpi.com/article/10.3390/min12030383/s1, Table S1: Whole-rock major oxides (wt.%) and trace elements (ppm) of the Kangjiali (047) gneissic granite and the Liwanxincun (045) metabasalt, Table S2: Whole-rock Sr and Nd isotopic data of the Kangjiali (047) gneissic granite and the Liwanxincun (045) metabasalt, Table S3: LA-MC-ICP-MS U-Pb isotopic data of zircons from the Sunjiaxia (068, 069) gneissic granite.

**Author Contributions:** Conceptualization, D.H.; methodology, D.H.; software, Z.L.; validation, Z.L., Y.X. and D.H.; formal analysis, Z.L. and Y.X.; investigation, D.H., Z.L. and Y.X.; resources, D.H.; data curation, Z.L.; writing—original draft preparation, Z.L.; writing—review and editing, D.H., W.X. and C.L.; visualization, Z.L.; supervision, D.H.; project administration, D.H.; funding acquisition, D.H. and C.L. All authors have read and agreed to the published version of the manuscript.

**Funding:** This research was funded by the National Natural Science Foundation of China, grant number 42130810, and the funding from project of Institute of Geology Chinese Academy of Geological Sciences, grant number 1212011120157.

**Data Availability Statement:** All data generated or used during the study appear in the submitted article.

**Acknowledgments:** We would like to thank the Wuhan Institute of Geology and Resources, Wuhan, China for their work in obtaining the major and trace elements, Sr- and Nd isotopes and the zircon U-Pb dating data. We are grateful to the anonymous reviewer for valuable comments and suggestions.

**Conflicts of Interest:** The authors declare no conflict of interest.

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
