# Peer review of "The Early Paleozoic Tectonic Framework and Evolution of Northern West Qinling Orogen: By Zircon U-Pb Dating and Geochemistry of Rocks from Tianshui and Sunjiaxia"

_minerals, doi:10.3390/min12030383_

Round 1

Reviewer 1 Report

See attached file.

Author Response

Thank you very much for your constructive comments and suggestions. They are very helpful to improve the quality manuscript and study. I have carefully addressed the comments and suggestions point-by-point. Please see the attachment.

Reviewer 2 Report

The reviewed manuscript describes integrated geochemical, isotope-geochemical and geochronological data for set of granitoids and metabasalt from the Qilian Orogen (Chinese Central Orogenic Belt). Based on these results authors made constrains the emplacement ages and petrogenesis of these rocks and provided tectonic implications of igneous rocks from the studied domain. The manuscript is generally clear, quite well written; the conclusions are mostly well supported by the results.

I have a minor comment on the isotope data presentation.

  1. Section 3.2. (Whole-rock Sr and Nd isotope analyses). This section contains only limited information about the procedure of isotope analyses, no Sr and Nd standards values as well as Sr chemical procedure described. Please, provide appropriate description.
  2. Calculation of TDM(Ga) for samples with 147Sm/144Nd>0.14 like in metabasalts samples (Table 2 in Supplementary files) is meaningless and incorrectly, please, remove these values from the Table. Besides, you need to reduce a number of signific digits in εNd(t) values – usually reported up to one decimal point.
  3. I would suggest to make a U-Pb data reduction used to age calculation. It would be more correct if you will use data with age discordance <10% for weighted age calculations. Your data will still representative (about 17 analyses for sample meet this criteria) but will much more evident. Also, please, add the D, % in the Table 3.

In summary, I support the publication of this paper with moderate revisions.

Author Response

Thank you very much for your feedback and suggestions. They are very helpful to improve the quality manuscript and study.  I have carefully addressed the comments and suggestions point-by-point. Please see the attachment.
